# COMPLETE AND EFFICIENT GRAPH TRANSFORMERS FOR CRYSTAL MATERIAL PROPERTY PREDICTION

**Keqiang Yan, Cong Fu, Xiaofeng Qian**[*]**, Xiaoning Qian**[*]**, Shuiwang Ji**[*]
Texas A&M University
College Station, TX 77843, USA
{keqiangyan,congfu,xqian,feng,sji}@tamu.edu

## ABSTRACT

Crystal structures are characterized by atomic bases within a primitive unit cell that repeats along a regular lattice throughout 3D space. The periodic and infinite nature of crystals poses unique challenges for geometric graph representation learning. Specifically, constructing graphs that effectively capture the complete geometric information of crystals and handle chiral crystals remains an unsolved and challenging problem. In this paper, we introduce a novel approach that utilizes the periodic patterns of unit cells to establish the lattice-based representation for each atom, enabling efficient and expressive graph representations of crystals. Furthermore, we propose ComFormer, a SE(3) transformer designed specifically for crystalline materials. ComFormer includes two variants; namely, iComFormer that employs invariant geometric descriptors of Euclidean distances and angles, and eComFormer that utilizes equivariant vector representations. Experimental results demonstrate the state-of-the-art predictive accuracy of ComFormer variants on various tasks across three widely-used crystal benchmarks. Our code is publicly available as part of the AIRS library (https://github.com/divelab/AIRS).

## 1 INTRODUCTION

Accelerating the discovery of novel materials with desirable properties has values in many science, engineering, and biomedical applications (Ramprasad et al., 2017; Meredig et al., 2014; Oliynyk et al., 2016; Raccuglia et al., 2016; Ward et al., 2016; Liu et al., 2021b;c; Luo et al., 2021; Xie et al., 2022; Fu et al., 2023; Wang et al.; Chanussot et al., 2021; Zhang et al., 2023). However, the current reliance on traditional, costly, and time-consuming trial-and-error experimental methods poses practical challenges. In this regard, computational approaches based on quantum mechanics, such as density functional theory (DFT), have made significant contributions for predicting the physical and chemical properties of materials to guide materials discovery experiments. Nevertheless, the high computational cost associated with these methods (ranging from $n^3$ to $n^7$, where $n$ represents the number of atoms) still poses a hurdle for novel materials discovery in the vast materials space.

To reduce computational cost, recent studies employ machine learning (ML) models (Yu et al., 2022; Ying et al., 2021; Gao & Ji, 2019; Liu et al., 2021a; Fu et al., 2022; Passaro & Zitnick, 2023; Gasteiger et al., 2020; Liao & Smidt, 2022; Batzner et al., 2022; Satorras et al., 2021; Qiao et al., 2022; Watson et al., 2022; Ahmad et al., 2018), such as crystal graph representations (Xie & Grossman, 2018; Ling et al., 2023) with deep neural networks. However, these crystal graph representations have limitations in distinguishing different crystalline materials. In other words, they cannot guarantee to capture the complete geometric information of input crystal structures, and may map different crystal structures with different properties to the same graph representation and produce the identical property predictions. Illustrative examples and detailed discussions can be found in Appendix A.1.

While graph representations that can capture any structural differences in small molecules have been investigated in previous works (Wang et al., 2022; Klicpera et al., 2021), these methods fail short to capture periodic patterns of crystals and cannot maintain geometric completeness for crystals. Thus, how to construct graph representations for crystalline materials that can distinguish any different crystals and remain the same under crystal passive symmetries detailed in Sec. 3.1 remains unsolved.

---

[*]Equal senior authorship

Establishing invariant (Anosova & Kurlin, 2022; Edelsbrunner et al., 2021) and complete representations have also been investigated for other applications, especially representations of matrix forms or consisting of aligned coordinates that can be used to measure the structural differences between crystals. Among them, AMD (Widdowson et al., 2022) and PDD (Widdowson & Kurlin, 2022) are two recent works that satisfy continuity under small perturbations of atom positions by using Euclidean distances. However, AMD and PDD cannot distinguish chiral crystals and cannot be directly used to predict crystal properties without breaking the completeness.

In this work, we focus on crystal property prediction and propose SE(3) invariant and SO(3) equivariant crystal graph representations to strive for geometric completeness for crystalline materials. Based on these expressive crystal graphs, we develop ComFormer with two variants; namely iComFormer that employs SE(3) invariant crystal graphs, and eComFormer that uses SO(3) equivariant crystal graphs. Both variants can scale to large-scale crystal datasets with complexity $O(nk)$ where $n$ denotes the number of atoms in the crystal unit cell and $k$ denotes the average number of neighbors. We evaluate the proposed ComFormer variants on three widely-used crystalline material benchmarks. The state-of-the-art performances of ComFormer variants on various tasks across different scales demonstrate the effectiveness of the proposed crystal graphs and ComFormer variants.

## 2 BACKGROUND AND RELATED WORK

### 2.1 CRYSTAL STRUCTURES AND CRYSTAL PROPERTY PREDICTION

Crystal structures consist of a unit cell and a set of atom bases associated with the unit cell. The unit cell is defined by a $3 \times 3$ lattice matrix indicating how the unit cell repeats itself in 3D space. Following the notations in Yan et al. (2022), crystal structures can be represented by $\mathbf{M} = (\mathbf{A}, \mathbf{P}, \mathbf{L})$, where $\mathbf{A} = [\boldsymbol{a}_1, \boldsymbol{a}_2, \cdots, \boldsymbol{a}_n] \in \mathbb{R}^{d_a \times n}$ contains the $d_a$-dimensional feature vectors for $n$ atoms in the unit cell, $\mathbf{P} = [\boldsymbol{p}_1, \boldsymbol{p}_2, \cdots, \boldsymbol{p}_n] \in \mathbb{R}^{3 \times n}$ contains the 3D Euclidean positions of $n$ atoms in the unit cell, and the lattice matrix $\mathbf{L} = [\boldsymbol{\ell}_1, \boldsymbol{\ell}_2, \boldsymbol{\ell}_3] \in \mathbb{R}^{3 \times 3}$ describes the repeating patterns of the unit cell structure in 3D space. Different from molecules and proteins, crystal structures naturally repeat infinitely in space. Formally, the infinite crystal structure can be described as

$$
\begin{aligned}
\hat{\mathbf{P}} &= \{\hat{\boldsymbol{p}}_i | \hat{\boldsymbol{p}}_i = \boldsymbol{p}_i + k_1 \boldsymbol{\ell}_1 + k_2 \boldsymbol{\ell}_2 + k_3 \boldsymbol{\ell}_3, \ k_1, k_2, k_3 \in \mathbb{Z}, i \in \mathbb{Z}, 1 \le i \le n\}, \\
\hat{\mathbf{A}} &= \{\hat{\boldsymbol{a}}_i | \hat{\boldsymbol{a}}_i = \boldsymbol{a}_i, i \in \mathbb{Z}, 1 \le i \le n\},
\end{aligned}
\tag{1}
$$

where the set $\hat{\mathbf{P}}$ represents all 3D positions for every atom $i$ in the repeated unit cell structure, and the set $\hat{\mathbf{A}}$ represents the corresponding atom feature vector for every atom. In this paper, we aim to predict the target property value $y$, where $y \in \mathbb{R}$ for regression tasks or $y \in \{1, 2, \cdots, C\}$ for classification tasks with $C$ classes by using crystal structure $(\mathbf{A}, \mathbf{P}, \mathbf{L})$ as input.

### 2.2 GEOMETRIC COMPLETENESS FOR CRYSTAL GRAPHS AND CHALLENGES

**Definition 1** (Geometrically Complete Crystal Graph)**.** *Following Widdowson & Kurlin (2022), a crystal graph $\mathcal{G}$ is geometrically complete if $\mathcal{G}_1 = \mathcal{G}_2 \rightarrow \mathbf{M}_1 \cong \mathbf{M}_2$, where $\cong$ denotes that two crystals are isometric as defined in Appendix A.3. It means if two crystal graphs $\mathcal{G}_1$ and $\mathcal{G}_2$ are the same, the infinite crystal structures represented by $\mathcal{G}_1$ and $\mathcal{G}_2$ are identical.*

A crystal graph $\mathcal{G}$ is considered geometrically complete if it can distinguish any minor structural differences between different crystalline materials.

**Challenges.** Achieving geometric completeness for crystals is more challenging than small molecules. Different from molecules, crystals consist of unit cell structures and corresponding periodic patterns, with infinite number of atom positions described by $\hat{\mathbf{P}} = \{\hat{\boldsymbol{p}}_i | \hat{\boldsymbol{p}}_i = \boldsymbol{p}_i + k_1 \boldsymbol{\ell}_1 + k_2 \boldsymbol{\ell}_2 + k_3 \boldsymbol{\ell}_3, \ k_1, k_2, k_3 \in \mathbb{Z}, i \in \mathbb{Z}, 1 \le i \le n\}$. Previous methods for molecular representation learning (Liu et al., 2022; Klicpera et al., 2021) only consider atom interactions within a radius, and cannot capture the periodic patterns of crystals. As a result, these methods fall short to maintain geometric completeness when extended to crystals as shown in Appendix A.1. Furthermore, different from rotation and translation invariances for molecular representation learning, unique crystal passive symmetries arisen from enormous number of choices of unit cell structures and corresponding periodic patterns to describe a crystal structure need to be considered, including unit cell SE(3) invariance, unit cell SO(3) equivariance, and periodic invariance as detailed in Sec. 3.1.

## 2.3 RELATED WORKS

**Crystal graph neural networks**. To capture atom interactions across cell boundaries, Xie & Grossman (2018) proposed CGCNN with multi-edge crystal graphs which use Euclidean distances as edge features. Based on the multi-edge crystal graph, Chen et al. (2019) and Louis et al. (2020) proposed different model architectures to enhance property prediction accuracy. In addition to Euclidean distances, Choudhary & DeCost (2021) and Chen & Ong (2022) proposed to include angle information and build line graphs with complexity $O(nk^2)$ to increase the geometric modeling capacity, where $n$ denotes the number of atoms in the unit cell and $k$ denotes the average number of neighbors for every atom. Recently, Matformer (Yan et al., 2022) was proposed to encode periodic patterns by self-connecting edges, and PotNet (Lin et al., 2023) considers the infinite summations of pairwise atomic interactions with additional computational cost $O(n^2)$. Despite the successes of previous crystal neural networks, it remains unsolved to achieve geometric completeness for crystalline materials, with demonstrations provided in Appendix A.1

**Geometric completeness for molecules**. To pursue geometric completeness for molecules, various approaches have been developed. SphereNet (Liu et al., 2022) achieves geometric completeness for molecules in most cases by using the combination of Euclidean distances, bond angles, and torsional angles, with some failure cases. GemNet (Klicpera et al., 2021) goes a step further by incorporating dihedral angles and achieving completeness for molecules with a complexity of $O(nk^3)$. ComENet (Wang et al., 2022) decreases the modeling complexity by selecting reference nodes using Euclidean distances. However, these methods fall short to capture periodic patterns (Yan et al., 2022) of crystals and are not geometrically complete for crystalline materials. As another branch of work, Graphormer (Ying et al., 2021) uses fully-connected molecule graphs and achieves completeness for molecules, but breaks periodic invariance as shown in Appendix A.1.2.

**Complete representations for crystals**. Constructing complete crystal representations in other forms is not a new topic. There are several recent efforts, including AMD (Widdowson et al., 2022) and PDD (Widdowson & Kurlin, 2022) of matrix forms that are complete. However, AMD and PDD cannot distinguish chiral crystals with different properties. Additionally, using these matrix form representations as input to predict crystal properties without breaking the completeness is challenging in practice. First, AMD and PDD representations are designed for stable crystal structures and do not consider atom types. The completeness of AMD and PDD is based on the assumption that no two crystals can have the same structure but differ by a single atom type, which is only feasible for stable crystal structures. Second, to achieve completeness, a sufficiently large number of neighbors $k$ needs to be predetermined for any test crystal (must be larger than the number of atoms in the cell for any test crystal), which is not feasible and very expensive as mentioned in PDD-ML (Balasingham et al., 2022), with more details provided in Appendix A.1.3. It is worth noting that our proposed graph representations can apply to dynamic crystal systems and are robust to various cell sizes.

## 3 GEOMETRICALLY COMPLETE CRYSTAL GRAPH REPRESENTATIONS

In this section, we first demonstrate crystal passive symmetries that will not change crystal structures, including unit cell SE(3) invariance, unit cell SO(3) equivariance, and periodic invariance. We discuss consequences when these symmetries are not encoded. We then propose SE(3) invariant and SO(3) equivariant crystal graph representations that achieve geometric completeness for crystalline materials. We further provide proofs that the proposed crystal graphs satisfy crystal passive symmetries in Appendix A.2, and are geometrically complete in Sec. 3.4, with verification provided in Appendix A.4.

### 3.1 CRYSTAL PASSIVE SYMMETRIES

Crystal structures are different from molecules and proteins and exhibit unique passive symmetries (Villar et al., 2023), including unit cell SE(3) invariance, unit cell SO(3) equivariance, and periodic invariance. Specifically, crystal structures remain unchanged when the position matrix $\mathbf{P}$ of the unit cell structure is translated alone or rotated together with $\mathbf{L}$, as shown in Fig. 1(b) and (c). Additionally, different minimum repeatable structures can be used to represent the same crystal as the periodic transformations shown in Fig. 1(d), and these different crystal structure representations ($\mathbf{A}$, $\mathbf{P}$, $\mathbf{L}$) introduce a crystal passive symmetry called periodic invariance. However, reflection

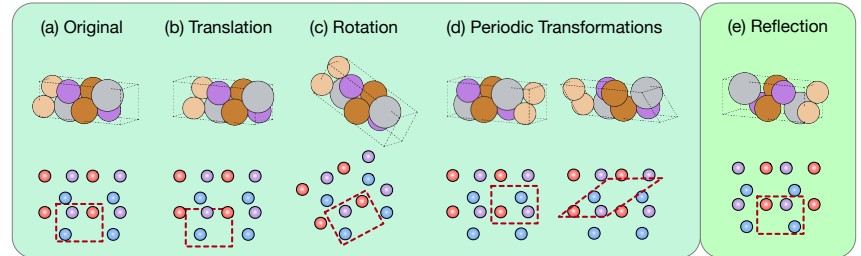

Figure 1: Illustrations of crystal passive symmetries. We show examples of a real crystal structure in 3D and corresponding simpler demonstrations in 2D. For the 2D demonstrations, we use circles with different colors to represent different kinds of atoms, and use red dotted lines to represent periodic unit cells. We show two cases of periodic transformations, including shifting the periodic unit cells and changing periodic unit cells without changing the volume. Translation, rotation, and periodic transformations will not change the crystal structure and are passive symmetries, while reflection transformation will map the crystal structure to its chiral image if the reflection symmetry is absent.

transformation is not a crystal passive symmetry as it maps a crystal structure to its chiral image. We then provide formal definitions of these crystal passive symmetries as follows.

**Definition 2** (Unit Cell SE(3) Invariance). *A function $f : (\mathbf{A}, \mathbf{P}, \mathbf{L}) \to \mathcal{X}$ is unit cell SE(3) invariant if, for any rotation transformation $\mathbf{R} \in \mathbb{R}^{3 \times 3}, |\mathbf{R}| = 1$ and translation transformation $b \in \mathbb{R}^3$, we have $f(\mathbf{A}, \mathbf{P}, \mathbf{L}) = f(\mathbf{A}, \mathbf{RP} + b, \mathbf{RL})$.*

A similar definition of unit cell E(3) invariance has been given in Yan et al. (2022). The major difference between unit cell E(3) and SE(3) invariance is that unit cell SE(3) invariant functions further distinguish crystal structures with different chiralities.

**Definition 3** (Unit Cell SO(3) Equivariance). *A function $f : (\mathbf{A}, \mathbf{P}, \mathbf{L}) \to \mathcal{X}' \in R^3$ is unit cell SO(3) equivariant if, for any rotation transformation $\mathbf{R} \in \mathbb{R}^{3 \times 3}, |\mathbf{R}| = 1$ and translation transformation $b \in \mathbb{R}^3$, we have $\mathbf{R} f(\mathbf{A}, \mathbf{P}, \mathbf{L}) = f(\mathbf{A}, \mathbf{RP} + b, \mathbf{RL})$.*

For a given infinite crystal structure $(\hat{\mathbf{P}}, \hat{\mathbf{A}})$ where $\hat{\mathbf{P}} = \{\hat{\boldsymbol{p}}_i | \hat{\boldsymbol{p}}_i = \boldsymbol{p}_i + k_1 \boldsymbol{\ell}_1 + k_2 \boldsymbol{\ell}_2 + k_3 \boldsymbol{\ell}_3, \ k_1, k_2, k_3 \in \mathbb{Z}, i \in \mathbb{Z}, 1 \le i \le n\}$ and $\hat{\mathbf{A}} = \{\hat{\boldsymbol{a}}_i | \hat{\boldsymbol{a}}_i = \boldsymbol{a}_i, i \in \mathbb{Z}, 1 \le i \le n\}$, periodic invariance describes that the representation of crystals should be invariant to the choice of minimum unit cell representations. The definition is detailed below.

**Definition 4** (Periodic Invariance). *A function $f : (\mathbf{A}, \mathbf{P}, \mathbf{L}) \to \mathcal{X}$ is periodic invariant if, for any possible minimum unit cell representations $\mathbf{M}' = (\mathbf{A}', \mathbf{P}', \mathbf{L}')$ representing a given infinite crystal structure $(\hat{\mathbf{P}}, \hat{\mathbf{A}})$, we have $f(\mathbf{A}, \mathbf{P}, \mathbf{L}) = f(\mathbf{A}', \mathbf{P}', \mathbf{L}')$.*

Two periodic transformations that can generate different minimum unit cell representations for the same crystal structure are shown in Fig. 1(d), including shifting periodic boundaries shown in the left and changing periodic patterns with the same unit cell volume shown in the right.

## 3.2 SE(3) INVARIANT CRYSTAL GRAPH REPRESENTATIONS

To achieve geometric completeness for crystalline materials, we introduce three periodic vectors, $\mathbf{e}_{\mathbf{ii}_1}$, $\mathbf{e}_{\mathbf{ii}_2}$, and $\mathbf{e}_{\mathbf{ii}_3}$, to serve as the lattice representation for node $i$. The lattice representations for different nodes are perfectly aligned because all atoms in a crystal share the same periodic patterns.

**Invariant crystal graph construction**. Every node in the proposed SE(3) invariant crystal graph represents atom $i$ and all its infinite duplicates in 3D space, with positions $\{\hat{\boldsymbol{p}}_i | \hat{\boldsymbol{p}}_i = \boldsymbol{p}_i + k_1 \boldsymbol{\ell}_1 + k_2 \boldsymbol{\ell}_2 + k_3 \boldsymbol{\ell}_3, \ k_1, k_2, k_3 \in \mathbb{Z}\}$ and node feature $\boldsymbol{a}_i$. An edge will be built from node $j$ to node $i$ when the Euclidean distance $||\mathbf{e}_{\mathbf{j'i}}||_2$ between a duplicate of $j$ and $i$ satisfies $||\mathbf{e}_{\mathbf{j'i}}||_2 = ||\boldsymbol{p}_j + k'_1 \boldsymbol{\ell}_1 + k'_2 \boldsymbol{\ell}_2 + k'_3 \boldsymbol{\ell}_3 - \boldsymbol{p}_i||_2 \le r$, where $r \in \mathbb{R}$ is the cutoff radius determined by the $k$-th nearest neighbor. Each edge has a corresponding feature vector $\{||\mathbf{e}_{\mathbf{j'i}}||_2, \theta_{j'i,ii_1}, \theta_{j'i,ii_2}, \theta_{j'i,ii_3}\}$, with $\theta_{j'i,ii_1}, \theta_{j'i,ii_2}$, and $\theta_{j'i,ii_3}$ denoting the angles between $\mathbf{e}_{\mathbf{j'i}}$ and lattice representation $\{\mathbf{e}_{\mathbf{ii}_1}, \mathbf{e}_{\mathbf{ii}_2}, \mathbf{e}_{\mathbf{ii}_3}\}$, respectively. Duplicates $i_1$, $i_2$, and $i_3$ are included as neighbors of $i$ to encode periodic patterns. The construction of the lattice representation $\{\mathbf{e}_{\mathbf{ii}_1}, \mathbf{e}_{\mathbf{ii}_2}, \mathbf{e}_{\mathbf{ii}_3}\}$ for node $i$ can be found below.

**Importance of periodic invariance for lattice representation**. A straightforward way to obtain lattice representation is $\{\mathbf{e_{ii_1}}, \mathbf{e_{ii_2}}, \mathbf{e_{ii_3}}\} = \{\boldsymbol{\ell}_1, \boldsymbol{\ell}_2, \boldsymbol{\ell}_3\}$, where $\boldsymbol{\ell}_1, \boldsymbol{\ell}_2, \boldsymbol{\ell}_3$ are the periodic lattice vectors provided in input lattice matrix $\mathbf{L}$. However, as shown in Sec. 3.1 and Fig. 1, the same crystal structure can have different lattice matrices $\mathbf{L}$ when applying periodic transformations. For instance, $\mathbf{L} = [\boldsymbol{\ell}_1, \boldsymbol{\ell}_2, \boldsymbol{\ell}_3]^T$ and $\mathbf{L'} = [\boldsymbol{\ell}_1 + \boldsymbol{\ell}_2, \boldsymbol{\ell}_2, \boldsymbol{\ell}_3]^T$ describe the same periodic patterns and have the same volume of the unit cell. Therefore, lattice representation needs to be invariant to periodic transformations.

**Lattice representation construction**. The lattice representation is constructed as follows to satisfy periodic invariance. First, we select the periodic duplicate $i_1$ of node $i$ that has the smallest edge length $||\mathbf{e_{ii_1}}||_2 \leq ||\mathbf{e_{ii'}}||_2$ for any other duplicate $i'$ where $\{\boldsymbol{p}'_i = \boldsymbol{p}_i + k_1\boldsymbol{\ell}_1 + k_2\boldsymbol{\ell}_2 + k_3\boldsymbol{\ell}_3, \; k_1, k_2, k_3 \in \mathbb{Z}\}$. We then select the periodic duplicate $i_2$ of node $i$ that has the smallest edge length $||\mathbf{e_{ii_2}}||_2 \leq ||\mathbf{e_{ii'}}||_2$ for any other duplicate $i'$ except $i_1$ and verify that $\mathbf{e_{ii_1}}$ and $\mathbf{e_{ii_2}}$ are not in the same line. If they are in the same line in 3D space, we select the duplicate with the next smallest distance and repeat until $\mathbf{e_{ii_2}}$ is found. We then adjust the vector $\mathbf{e_{ii_2}}$ to $-\mathbf{e_{ii_2}}$ if the relative angle between $\mathbf{e_{ii_2}}$ and $\mathbf{e_{ii_1}}$ is larger than 90°. Third, we select the duplicate with the next smallest distance $||\mathbf{e_{ii_3}}||_2$, and verify that $\mathbf{e_{ii_3}}, \mathbf{e_{ii_1}}$ and $\mathbf{e_{ii_2}}$ are not in the same plane in 3D space, and repeat until $\mathbf{e_{ii_3}}$ is found, then adjust the vector $\mathbf{e_{ii_3}}$ to $-\mathbf{e_{ii_3}}$ if the relative angle between $\mathbf{e_{ii_3}}$ and $\mathbf{e_{ii_1}}$ is larger than 90°. In the end, we will verify that $\{\mathbf{e_{ii_1}}, \mathbf{e_{ii_2}}, \mathbf{e_{ii_3}}\}$ forms a right-hand system, if not, convert them by $\{-\mathbf{e_{ii_1}}, -\mathbf{e_{ii_2}}, -\mathbf{e_{ii_3}}\}$ to form a right-hand system. The proposed lattice representation $\{\mathbf{e_{ii_1}}, \mathbf{e_{ii_2}}, \mathbf{e_{ii_3}}\}$ satisfies the unit cell SO(3) equivariance and periodic invariance, with detailed proofs provided in Appendix A.2.1. We also provide proofs in Appendix A.2.1 that it distinguishes different chiral crystal structures.

### 3.3 SO(3) EQUIVARIANT CRYSTAL GRAPH REPRESENTATIONS

In addition to SE(3) invariant crystal graphs, we propose SO(3) equivariant crystal graphs that achieve geometric completeness by using vector representations.

**Equivariant crystal graph construction**. The only difference between the proposed equivariant crystal graphs and invariant crystal graphs is the edge feature. Every edge in the equivariant crystal graph has the invariant edge feature $||\mathbf{e_{j'i}}||_2$ and the equivariant edge feature $\mathbf{e_{j'i}} \in \mathbb{R}^3$.

With 3D vectors as edge features, the unit cell SO(3) equivariant property can be achieved as proven in Appendix A.2.2 . However, SO(3) equivariant crystal graphs pose challenges to consider expressive geometric information during message passing. If only the norm of 3D vectors were used during message passing, it will be the same as previous methods using Eucldiean distances only, such as SchNet (Schütt et al., 2017) and Matformer (Yan et al., 2022).

### 3.4 GEOMETRIC COMPLETENESS OF PROPOSED CRYSTAL GRAPHS

**Proposition 1.** *The SE(3) invariant and SO(3) equivariant crystal graphs are geometrically complete.*

*Proof.* We prove by mathematical induction. We assume that the number of atoms (nodes) in the crystal unit cell is $n$. Note that the crystal graph we consider is strongly connected, which means that there exists a path between any two nodes in the graph. All crystalline materials in nature can be constructed as strongly connected graphs.

Base case: The infinite crystal structure represented by SE(3) invariant or SO(3) equivariant crystal graphs is unique when $n = 1$.

Inductive hypothesis: The infinite crystal structure is unique holds for $n$ up to $m \geq 1$.

Inductive step: Let $n = m + 1$. Without loss of generality, among the existing $m$ nodes, we safely assume $\mathcal{N}_j$ is the set of nodes that form the local region of node $j$. Then $j$ is the index of the $(m + 1)$-th node newly connected to these nodes. To prove the infinite crystal structure is still unique, we only need to prove that the relative position of node $j$ is uniquely determined given the SE(3) invariant crystal graph or the SO(3) equivariant crystal graph. The proofs of Base and Inductive cases are provided in Appendix A.3. With that, the proposed SE(3) invariant and SO(3) equivariant crystal graphs can determine a unique infinite crystal structure. Hence, the same crystal graphs proposed will only represent the identical infinite crystal structure. Then, based on Definition 1, we complete the proof of Proposition 1. $\square$

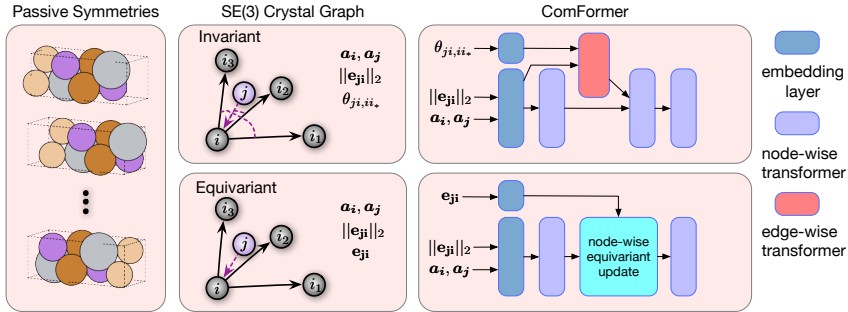

Figure 2: Illustration of the proposed ComFormer pipeline. In the left figure, we show different unit cell structures for the same crystal due to passive crystal symmetries, and all of them will map to the same invariant or equivariant crystal graph shown in the middle. In the middle, we demonstrate the information included in our proposed SE(3) invariant and equivariant crystal graphs. Specifically, we include node feature $\boldsymbol{a}_i$ for every node $i$, and for every neighbor $j$ of node $i$, we include edge length $||\mathbf{e_{ji}}||_2$, and three angles $\theta_{ji,ii_*} = \{\theta_{ji,ii_1}, \theta_{ji,ii_2}, \theta_{ji,ii_3}\}$ in invariant crystal graphs, and edge vector $\mathbf{e_{ji}}$ in equivariant crystal graphs. The proposed iComFormer and eComFormer are shown on the right, with building blocks marked by different colors.

## 4 EFFICIENT AND EXPRESSIVE CRYSTAL TRANSFORMERS

We then propose iComFormer and eComFormer for crystal representation learning, based on the proposed SE(3) invariant and SO(3) equivariant crystal graphs in an efficient manner.

### 4.1 SE(3) INVARIANT MESSAGE PASSING

In this section, we first present iComFormer as shown in the upper right panel of Fig. 2. iComFormer utilizes SE(3) invariant crystal graphs with computational complexity of $O(nk)$, where $n$ is the number of nodes, and $k$ is the average number of neighbors for every node.

**Node and edge feature embedding layer**. Node feature $\boldsymbol{a_i}$ for node $i$ is mapped to $\boldsymbol{f}_i^0$ using CGCNN (Xie & Grossman, 2018) embedding features as the initial node feature before transformer layers. Edge feature $||\mathbf{e_{ji}}||_2$ for node pair $j$ and $i$ is mapped to $c/||\mathbf{e_{ji}}||_2$ and embedded by RBF kernels to initial edge feature $\boldsymbol{f}_{ji}^e$, where $c$ is a chosen constant to mimic the pairwise potential calculation following Lin et al. (2023). Three angles $\theta_{ji,ii_1}$, $\theta_{ji,ii_2}$, and $\theta_{ji,ii_3}$ are mapped to $\boldsymbol{f}_{ji_1}^\theta$, $\boldsymbol{f}_{ji_2}^\theta$, and $\boldsymbol{f}_{ji_3}^\theta$ by cosine function and RBF kernels. More details are provided in Appendix A.6.

**Node-wise transformer layer**. Node-wise transformer layer first passes messages from a neighboring node $j$ to the center node $i$ using node features $\boldsymbol{f}_j^l$, $\boldsymbol{f}_i^l$ and edge feature $\boldsymbol{f}_{ji}^e$ where $l$ indicates the layer number, and then aggregates all neighboring messages to update $\boldsymbol{f}_i^l$. The message from node $j$ to $i$ is formed by the corresponding query $\boldsymbol{q}_{ji}$, key $\boldsymbol{k}_{ji}$, and value features $\boldsymbol{v}_{ji}$ as follows,

$$\boldsymbol{q}_{ji} = \text{LN}_Q(\boldsymbol{f}_i^l), \boldsymbol{k}_{ji} = (\text{LN}_K(\boldsymbol{f}_i^l)|\text{LN}_K(\boldsymbol{f}_j^l)|\text{LN}_E(\boldsymbol{f}_{ji}^e)), \boldsymbol{v}_{ji} = (\text{LN}_V(\boldsymbol{f}_i^l)|\text{LN}_V(\boldsymbol{f}_j^l)|\text{LN}_E(\boldsymbol{f}_{ji}^e))$$

$$\boldsymbol{\alpha}_{ji} = \frac{\boldsymbol{q}_{ji} \circ \boldsymbol{\sigma}_K(\boldsymbol{k}_{ji})}{\sqrt{d_{\boldsymbol{q}_{ji}}}}, \ \boldsymbol{msg}_{ji} = \text{sigmoid}(\text{BN}(\boldsymbol{\alpha}_{ji})) \circ \boldsymbol{\sigma}_V(\boldsymbol{v}_{ji}),$$

(2)

where $\text{LN}_Q, \text{LN}_K, \text{LN}_V, \text{LN}_E$ are the linear transformations for query, key, value, and edge features. $\boldsymbol{\sigma}_K, \boldsymbol{\sigma}_V$ are the nonlinear transformations for key and value, including two linear layers and an activation layer in between. $\circ$ and $|$ denote the Hadamard product and concatenation, respectively. BN denotes the batch normalization layer, and $d_{\boldsymbol{q}_{ji}}$ is the dimension of $\boldsymbol{q}_{ji}$. In our design, the query in the transformer layer only contains the center node feature, while the key and value contain the center node, neighboring node, and edge features. Then, node feature $\boldsymbol{f}_i^l$ is updated using aggregated messages from neighbors to center node $i$ as follows,

$$\boldsymbol{msg}_i = \sum_{j \in \mathcal{N}_i} \boldsymbol{msg}_{ji}, \ \boldsymbol{f}_i^{l+1} = \boldsymbol{\sigma}_{msg}(\boldsymbol{f}_i^l + \text{BN}(\boldsymbol{msg}_i)),$$

(3)

where $\boldsymbol{\sigma}_{msg}$ denotes the softplus activation function. Because we calculate $k$ messages for every node $i$ where $k$ denotes the average degree, the complexity of node-wise transformer layer is $O(nk)$.

**Edge-wise transformer layer**. Edge-wise transformer layer updates edge feature $\boldsymbol{f}_{ji}^e$ using angle features $\boldsymbol{f}_{ji_1}^\theta, \boldsymbol{f}_{ji_2}^\theta$, and $\boldsymbol{f}_{ji_3}^\theta$, and edge features of lattice vectors $\boldsymbol{f}_{ii_1}^e, \boldsymbol{f}_{ii_2}^e$, and $\boldsymbol{f}_{ii_3}^e$. Detailed updating computation of edge-wise transformer layer is as follows,

$$
\begin{aligned}
\boldsymbol{q}_{ji}^e =& \mathrm{LN}_Q^e(\boldsymbol{f}_{ji}^e), \boldsymbol{k}_{ji_m}^e = (\mathrm{LN}_K^e(\boldsymbol{f}_{ji}^e)|\mathrm{LN}_K^{\theta_m}(\boldsymbol{f}_{ii_m}^e)), \boldsymbol{v}_{ji_m}^e = (\mathrm{LN}_V^e(\boldsymbol{f}_{ji}^e)|\mathrm{LN}_V^{\theta_m}(\boldsymbol{f}_{ii_m}^e)), \\
\boldsymbol{\alpha}_{ji_m}^e =& \frac{\boldsymbol{q}_{ji}^e \circ \boldsymbol{\sigma}_K(\boldsymbol{k}_{ji_m}^e|\mathrm{LN}_\theta(\boldsymbol{f}_{ji_m}^\theta))}{\sqrt{d_{\boldsymbol{q}_{ji}^e}}}, \boldsymbol{msg}_{ji_m}^e = \mathrm{sigmoid}(\mathrm{BN}(\boldsymbol{\alpha}_{ji}^{e_m})) \circ \boldsymbol{\sigma}_V^e(\boldsymbol{v}_{ji_m}^e|\mathrm{LN}_\theta(\boldsymbol{f}_{ji_m}^\theta)),
\end{aligned}
\tag{4}
$$

where $\mathrm{LN}_Q^e, \mathrm{LN}_K^e, \mathrm{LN}_V^e$ represent linear transformations for edge feature $\boldsymbol{f}_{ji}^e$, $\mathrm{LN}_K^{\theta_m}, \mathrm{LN}_V^{\theta_m}, m \in \{1, 2, 3\}$ represent linear transformations for edge features of lattice vectors, and $\mathrm{LN}_\theta$ is the linear transformation for angle features $\boldsymbol{f}_{ji_1}^\theta, \boldsymbol{f}_{ji_2}^\theta, \boldsymbol{f}_{ji_3}^\theta$. Then, edge feature $\boldsymbol{f}_{ji}^e$ is updated using aggregated messages from three lattice vectors as below,

$$
\boldsymbol{msg}_{ji}^e = \sum_{m \in \{0,1,2\}} \boldsymbol{msg}_{ji_m}^e, \ \boldsymbol{f}_{ji}^{e*} = \boldsymbol{\sigma}_{msg}(\boldsymbol{f}_{ji}^e + \mathrm{BN}(\boldsymbol{msg}_{ji}^e)),
\tag{5}
$$

because every edge $ji$ has three neighbors, the complexity of edge-wise transformer layer is $O(3nk) = O(nk)$. Hence, the total complexity of iComFormer is $O(nk)$. The proof that iComFormer considers expressive geometric information in message passing is shown in Appendix A.5.

## 4.2 SO(3) EQUIVARIANT MESSAGE PASSING

In this section, we describe SO(3) equivariant message passing for eComFormer, as shown in the lower right panel of Fig. 2. For the design of eComFormer message passing, a core question to answer is how to capture expressive geometric information in SO(3) crystal graphs. Specifically, different from Euclidean distances, vector features $\mathbf{e_{ji}}$ do not support nonlinear mappings. To tackle this, we employ the recent advances in tensor product (TP) layers (Geiger & Smidt, 2022a) and capture the expressive geometric information by properly stacking several TP layers together.

**Node and edge feature embedding layer**. Node feature $\boldsymbol{a_i}$ and edge feature $||\mathbf{e_{ji}}||_2$ for node pair $j$ and $i$ are embedded in the same way as iComFormer. Equivariant edge feature $\mathbf{e_{ji}}$ is embedded by corresponding spherical harmonics $\mathbf{Y}_0(\hat{\mathbf{e_{ji}}}) = c_0, \mathbf{Y}_1(\hat{\mathbf{e_{ji}}}) = c_1 * \hat{\mathbf{e_{ji}}} \in \mathbb{R}^3$ and $\mathbf{Y}_2(\hat{\mathbf{e_{ji}}}) \in \mathbb{R}^5$ where $\hat{\mathbf{e_{ji}}} = \frac{\mathbf{e_{ji}}}{||\mathbf{e_{ji}}||_2}$ and $c_0, c_1$ are constants, with more details provided in Appendix A.6.

**Node-wise equivariant updating layer**. To capture geometric information represented by vector features $\mathbf{e_{ji}}$ in SO(3) crystal graphs, we propose the node-wise equivariant updating layer. Specifically, it implicitly considers expressive geometric information by stacking two tensor product layers and taking edge features $\{\mathbf{Y}_0(\hat{\mathbf{e_{ji}}}), \mathbf{Y}_1(\hat{\mathbf{e_{ji}}}), \mathbf{Y}_2(\hat{\mathbf{e_{ji}}})\}$ and node features $\boldsymbol{f}_i^l$ as input. The first tensor product layer aggregates higher rotation order neighborhood information to the center node $i$ as follows,

$$
\boldsymbol{f}_{i,0}^l = \boldsymbol{f}_i^{l'} + \frac{1}{|\mathcal{N}_i|} \sum_{j \in \mathcal{N}_i} \mathbf{TP}_0(\boldsymbol{f}_j^{l'}, \mathbf{Y}_0(\hat{\mathbf{e_{ji}}})), \boldsymbol{f}_{i,\lambda}^l = \frac{1}{|\mathcal{N}_i|} \sum_{j \in \mathcal{N}_i} \mathbf{TP}_\lambda(\boldsymbol{f}_j^{l'}, \mathbf{Y}_\lambda(\hat{\mathbf{e_{ji}}})), \lambda \in \{1, 2\}, \tag{6}
$$

where $\boldsymbol{f}_i^{l'} = \mathrm{LN}(\boldsymbol{f}_i^l)$, $|\mathcal{N}_i|$ is the number of neighbors of node $i$, and $\mathbf{TP}_0, \mathbf{TP}_1, \mathbf{TP}_2$ are tensor product layers that produce outputs with rotation order equal 0, 1, and 2, respectively. Then, the second tensor product layer produce SE(3) invariant node features as below,

$$
\begin{aligned}
\boldsymbol{f}_i^{l*} =& \frac{1}{|\mathcal{N}_i|}(\sum_{j \in \mathcal{N}_i} \mathbf{TP}_0(\boldsymbol{f}_{j,0}^l, \mathbf{Y}_0(\hat{\mathbf{e_{ji}}})) + \sum_{j \in \mathcal{N}_i} \mathbf{TP}_0(\boldsymbol{f}_{j,1}^l, \mathbf{Y}_1(\hat{\mathbf{e_{ji}}})) + \sum_{j \in \mathcal{N}_i} \mathbf{TP}_0(\boldsymbol{f}_{j,2}^l, \mathbf{Y}_2(\hat{\mathbf{e_{ji}}}))), \\
\boldsymbol{f}_{i,updated}^l =& \boldsymbol{\sigma}_{\mathrm{equi}}(\mathrm{BN}(\boldsymbol{f}_i^{l*})) + \mathrm{LN}_{\mathrm{equi}}(\boldsymbol{f}_i^l),
\end{aligned}
\tag{7}
$$

where $\boldsymbol{\sigma}_{\mathrm{equi}}$ is a nonlinear transformation consisting of two softplus layers and a linear layer in between. We then show that by the proper design of tensor product layers, the output $\boldsymbol{f}_{i,updated}^l$ of node-wise equivariant updating layer considers all two-hop bond angle information $\sum_{j \in \mathcal{N}_i} \sum_{m \in \mathcal{N}_j} \mathrm{COS}(\theta_{mj,ji})$ in $\sum_{j \in \mathcal{N}_i} \mathbf{TP}_0(\boldsymbol{f}_{j,1}^l, \mathbf{Y}_1(\hat{\mathbf{e_{ji}}}))$, with details provided in Appendix A.5.

## 5 EXPERIMENTS

We assess the expressiveness of our iComFormer and eComFormer models by conducting evaluations on three widely-used crystal benchmarks: JARVIS (Choudhary et al., 2020), the Materials Project (Chen et al., 2019), and MatBench (Dunn et al., 2020). Through extensive experiments, our proposed ComFormer exhibits excellent prediction accuracy across a range of crystal properties, underscoring its modeling capabilities for crystalline materials.

For JARVIS and MP, we follow the experimental settings of Matformer (Yan et al., 2022) and PotNet (Lin et al., 2023), and use mean absolute error (MAE) as the evaluation metric. Among these tasks, there are large scale tasks with 69239 crystals, medium scale tasks with 18171 crystals, and small scale tasks with 5450 crystals. To further evaluate the performances, we use e_form with 132752 crystals and jdft2d with only 636 2D crystals in MatBench. The baseline methods include CGCNN (Xie & Grossman, 2018), MEGNET (Chen et al., 2019), GATGNN (Louis et al., 2020), ALIGNN (Choudhary & DeCost, 2021), Matformer (Yan et al., 2022), PotNet (Lin et al., 2023), M3GNet (Chen & Ong, 2022), MODNet (De Breuck et al., 2021), and coGN (Ruff et al., 2023). For tasks in these three datasets, the best results are shown in **bold** and the second best results are shown with underlines. We use one TITAN A100 for computing. More details of ComFormer settings and benchmark details can be found in Appendix A.6 and A.7, respectively.

### 5.1 EXPERIMENTAL RESULTS

**JARVIS**. As shown in Table 1, compared with highly competitive baselines, ComFormer variants achieve the best performances on all tasks. Specifically, iComFormer achieves 8% improvement beyond PotNet on formation energy, 10% improvement on $E_{total}$, and 15% improvement on $E_{hull}$, while eComFormer achieves 20% improvement on $E_{hull}$ and the second best performances on other three tasks.

Table 1: Comparison on JARVIS in terms of test MAE.

| Method | Form. Energy meV/atom | Bandgap(OPT) eV | $E_{total}$ meV/atom | Bandgap(MBJ) eV | $E_{hull}$ meV |
|---|---|---|---|---|---|
| CGCNN | 63 | 0.20 | 78 | 0.41 | 170 |
| SchNet | 45 | 0.19 | 47 | 0.43 | 140 |
| MEGNET | 47 | 0.145 | 58 | 0.34 | 84 |
| GATGNN | 47 | 0.170 | 56 | 0.51 | 120 |
| ALIGNN | 33.1 | 0.142 | 37 | 0.31 | 76 |
| Matformer | 32.5 | 0.137 | 35 | 0.30 | 64 |
| PotNet | 29.4 | 0.127 | 32 | 0.27 | 55 |
| eComFormer | 28.4 | 0.124 | 32 | 0.28 | **44** |
| iComFormer | **27.2** | **0.122** | **28.8** | **0.26** | 47 |

**The Materials Project (MP)**. Experimental results on MP are shown in Table 2. It can be seen that iComFormer achieves the best performances on three out of four tasks with significant margins beyond previous works, while eComFormer achieves the best on formation energy. Specifically, the excellent prediction accuracy of iComFormer on bulk moduli and shear moduli with only 4664 training samples shows the expressiveness and robustness of the invariant crystal graphs with limited training samples.

Table 2: Comparison on The Materials Project.

| Method | Form. Energy meV/atom | Band Gap eV | Bulk Moduli log(GPa) | Shear Moduli log(GPa) |
|---|---|---|---|---|
| CGCNN | 31 | 0.292 | 0.047 | 0.077 |
| SchNet | 33 | 0.345 | 0.066 | 0.099 |
| MEGNET | 30 | 0.307 | 0.060 | 0.099 |
| GATGNN | 33 | 0.280 | 0.045 | 0.075 |
| ALIGNN | 22 | 0.218 | 0.051 | 0.078 |
| Matformer | 21.0 | 0.211 | 0.043 | 0.073 |
| PotNet | 18.8 | 0.204 | 0.040 | 0.0650 |
| eComFormer | **18.16** | 0.202 | 0.0417 | 0.0729 |
| iComFormer | 18.26 | **0.193** | **0.0380** | **0.0637** |

**MatBench**. Experimental results for MatBench are shown in Table 3 in terms of MAE and RMSE with standard deviation of five benchmark runs. For the largest scale e_form with more than 130K crystals, ComFormer variants achieve the state-of-the-art performances, shedding light on the elegant integration of angle information in the complete crystal representations beyond other methods using angle information like ALIGNN and M3GNet. For jdft2d task,

Table 3: Comparison on MatBench.

| Methods | e_form-132752 MAE | e_form-132752 RMSE | jdft2d-636 MAE | jdft2d-636 RMSE |
|---|---|---|---|---|
| MODNet | $44.8 \pm 3.9$ | $88.8 \pm 7.5$ | **$33.2 \pm 7.3$** | $96.7 \pm 40.4$ |
| ALIGNN | $21.5 \pm 0.5$ | $55.4 \pm 5.5$ | $43.4 \pm 8.9$ | $117.4 \pm 42.9$ |
| coGN | $17.0 \pm 0.3$ | $48.3 \pm 5.9$ | $37.2 \pm 13.7$ | $101.2 \pm 55.0$ |
| M3GNet | $19.5 \pm 0.2$ | - | $50.1 \pm 11.9$ | - |
| eComFormer | **$16.5 \pm 0.3$** | $45.4 \pm 4.7$ | $37.8 \pm 9.0$ | $102.2 \pm 46.4$ |
| iComFormer | **$16.5 \pm 0.3$** | **$43.8 \pm 3.7$** | $34.8 \pm 9.9$ | **$96.1 \pm 46.3$** |

iComFormer is competitive with MODNet, the state-of-the-art physical descriptor based method for materials, indicating the excellent modeling power of iComFormer with only hundreds training samples. Overall, the robust performances of ComFormer variants on these three benchmarks for

various crystal properties with different data scales demonstrate the effectiveness of the proposed invariant and equivariant crystal representations and ComFormer architectures.

**Efficiency**. We evaluate the model complexity and running time of ComFormer variants by comparing them with geometrically incomplete methods Matformer and ALIGNN on the JARVIS formation energy task. To make the comparison fair, we follow Matformer and ALIGNN and use $k = 12$ when constructing crystal graphs. Further analysis when increasing $k$ is provided in Appendix A.6.2. We use 3 and 4 to indicate the number of node layers

Table 4: Efficiency analysis.

| Method | Complexity | Num. Params. | Time/epoch |
|---|---|---|---|
| Matformer | $O(nk)$ | 2.9 M | 64 s |
| ALIGNN | $O(nk^2)$ | 4.0 M | 327 s |
| eComFormer | $O(nk)$ | 12.4 M | 115 s |
| eComFormer-half | $O(nk)$ | 5.6 M | 90 s |
| iComFormer(3) | $O(nk)$ | 4.1 M | 69 s |
| iComFormer(4) | $O(nk)$ | 5.0 M | 78 s |

of iComFormer, which covers all experimental settings of iComFormer. As shown in Table 4, our proposed ComFormer variants are significantly more efficient than ALIGNN and are comparable with Matformer which only uses Euclidean distances.

Table 5: Verification of geometric completeness of proposed SE(3) invariant and SO(3) equivariant crystal graphs by reconstructing crystal structures from graph representations.

| Structure RMSD | MP | T2 (Pulido et al., 2017) |
|---|---|---|
| SE(3) invariant | $3.28 * 10^{-7}$ | $3.83 * 10^{-7}$ |
| SO(3) equivariant | $3.03 * 10^{-7}$ | $3.19 * 10^{-7}$ |

**Completeness verification**. We evaluate the completeness of proposed SE(3) invariant and SO(3) equivariant crystal graphs by reconstructing crystal structures from graph representations. Specifically, we use crystals from the Materials Project and T2 dataset which contains super large crystal cells with more than 700 hundred atoms to demonstrate the robustness of these crystal graphs. The detailed reconstruction process is provided in Appendix. A.4. After reconstruction, we compare the structure differences between the recovered ones and the input ones using Pymatgen. As shown in Table. 5, the input crystal structures are fully recovered, with limited structure error (less than $10^{-6}$).

## 5.2 ABLATION STUDIES

In this section, we demonstrate the importance of geometric complete crystal graphs and Comformer components by conducting ablation studies on JARVIS formation energy task.

**Importance of geometric complete crystal graphs**. We evaluate the importance of geometric completeness by making the proposed crystal graphs incomplete. Specifically, we omit the edge-wise transformer layer in iComFormer and the equivariant updating layer in eComFormer, which means that we omit bond angles and bond vectors in corresponding graphs. As shown in Table 6, using the incomplete crystal graphs will

Table 6: Importance of completeness.

| Method | Complete | Test MAE |
|---|---|---|
| w/o bond vectors | ✗ | 30.2 |
| eComFormer | ✓ | **28.4** |
| w/o bond angles | ✗ | 28.9 |
| iComFormer | ✓ | **27.2** |

degrade the performance of iComFormer by $6.25\%$ from 0.0272 to 0.0289, and the performance of eComFormer by $6.33\%$ from 0.0284 to 0.0302.

## 6 CONCLUSION, LIMITATIONS, AND FUTURE WORKS

In this work, we propose SE(3) ComFormer variants for crystalline materials that utilize geometrically complete crystal graphs and satisfy crystal passive symmetries. The state-of-the-art performances of ComFormer variants on various tasks across three widely-used crystal benchmarks demonstrate the expressive power of proposed geometrically complete crystal graphs and modeling capacity of ComFormer variants. The limitations of our current method include (1) it is designed for crystalline materials and cannot be directly applied to other domains, (2) it cannot model the complete geometric information of amorphous materials, and (3) it currently cannot predict higher order crystal properties. We plan to explore these directions in the future work. We also include discussions of potential corner cases in Appendix A.8 to aid future users of the proposed method in need.

ACKNOWLEDGMENTS

We thank Youzhi Luo, Limei Wang, and Haiyang Yu for insightful discussions, and thank Chengkai Liu for drawing 3D visualizations of crystal structures. S.J. acknowledges the support from National Science Foundation grants IIS-2243850 and CNS-2328395. XF Q. acknowledges the support from the Center for Reconfigurable Electronic Materials Inspired by Nonlinear Dynamics (reMIND), an Energy Frontier Research Center funded by the U.S. Department of Energy, Basic Energy Sciences, under Award Number DE-SC0023353. XN Q. acknowledges the support from National Science Foundation grants CCF-1553281, DMR-2119103, and IIS-2212419.

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

# A APPENDIX

## A.1 LIMITATIONS OF PREVIOUS METHODS IN ACHIEVING GEOMETRIC COMPLETENESS FOR CRYSTALLINE MATERIALS

### A.1.1 LIMITATIONS OF PREVIOUS CRYSTAL GRAPH NEURAL NETWORKS

Crystalline materials are first converted into graph representations which serve as the input of crystal graph neural networks. Thus, the geometric information considered by the crystal graph neural networks are directly determined by the employed crystal graphs.

In this subsection, we discuss the **limitations** of previous crystal graph neural networks in terms of crystal graph representations. These methods include CGCNN (Xie & Grossman, 2018), SchNet (Schütt et al., 2017), MEGNET (Chen et al., 2019), and GATGNN (Louis et al., 2020) which directly use multi-edge crystal graphs, ALIGNN (Choudhary & DeCost, 2021) and M3GNet (Chen & Ong, 2022) which uses multi-edge crystal graphs with extra bond angles, and Matformer (Yan et al., 2022) and PotNet (Lin et al., 2023) which use additional edge features to capture periodic patterns.

We first recap multi-edge crystal graphs, and then show the limitations of previous works in distinguishing different crystalline materials.

**Multi-edge crystal graphs**. The multi-edge crystal graph is proposed by Xie & Grossman (2018) to capture atomic interactions across periodic boundaries. As shown in Fig. 3, every node in the multi-edge crystal graph represents atom $i$ and all its infinite duplicates in 3D space, with positions $\{\hat{\boldsymbol{p}}_i | \hat{\boldsymbol{p}}_i = \boldsymbol{p}_i + k_1 \boldsymbol{\ell}_1 + k_2 \boldsymbol{\ell}_2 + k_3 \boldsymbol{\ell}_3, \ k_1, k_2, k_3 \in \mathbb{Z}\}$ and node feature $\boldsymbol{a}_i$. An edge will be built from node $j$ to node $i$ when the Euclidean distance $||\mathbf{e}_{\mathbf{j'i}}||_2$ between a duplicate of $j$ and $i$ satisfies $||\mathbf{e}_{\mathbf{j'i}}||_2 = ||\boldsymbol{p}_j + k_1^{'} \boldsymbol{\ell}_1 + k_2^{'} \boldsymbol{\ell}_2 + k_3^{'} \boldsymbol{\ell}_3 - \boldsymbol{p}_i||_2 \leq r$, where $r \in \mathbb{R}$ is the cutoff radius. Each edge has a corresponding edge feature $||\mathbf{e}_{\mathbf{j'i}}||_2$.

**Limitations of multi-edge crystal graphs**. (1) It can be seen from Fig. 4 that multi-edge crystal graphs **fail short to distinguish crystals with different periodic patterns**, and may map different crystals with different properties to the same graph. (2) It can be seen from Fig. 5 that multi-edge crystal graphs **fail short to distinguish crystalline materials with different unit cell structures**,

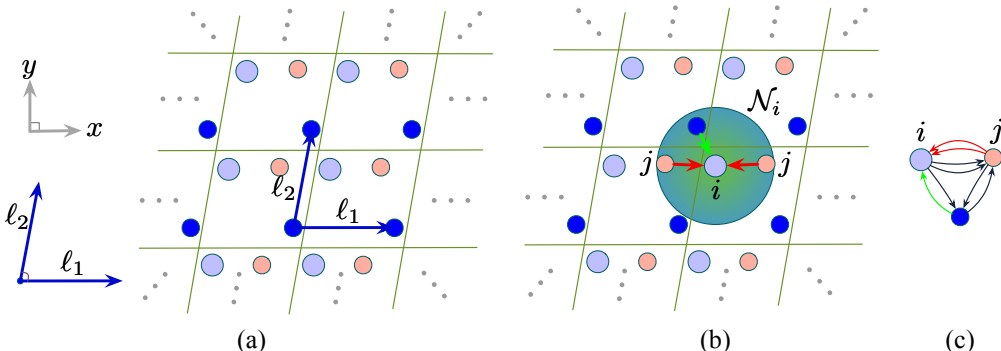

Figure 3: Demonstration of the multi-edge crystal graph. We use blue arrows to represent periodic patterns $\ell_1$ and $\ell_2$ for the shown crystal structure in 2D case. We use circles with different colors to represent different atoms, and light-green lines to represent the periodic boundaries. We use $\mathcal{N}_i$ to represent the neighborhood of node $i$ within the radius, and use red and green arrows to indicate the captured atomic interactions by the multi-edge crystal graph. (a) A crystal structure with periodic patterns $\ell_1$ and $\ell_2$ in 2D case for clarity. (b) The captured atomic interactions by the multi-edge crystal graph, between the red nodes $j$ and center node $i$. (c) The corresponding multi-edge crystal graph. All periodic duplicates of $j$ are mapped to a single node $j$ in the multi-edge crystal graph.

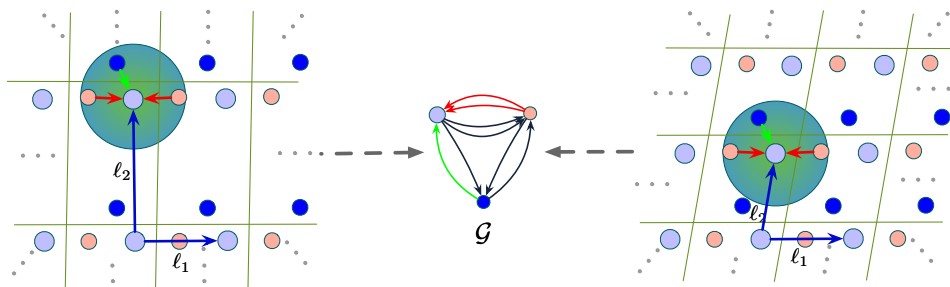

Figure 4: Demonstration that multi-edge crystal graphs fail short to capture periodic patterns. We use blue arrows to represent periodic patterns $\ell_1$ and $\ell_2$ for these two different crystal structures. Due to the missing geometric information of periodic patterns, multi-edge crystal graphs map these two different crystals to the same crystal graph $\mathcal{G}$.

and may map different crystals with different properties to the same graph. Thus, CGCNN (Xie & Grossman, 2018), SchNet (Schütt et al., 2017), MEGNET (Chen et al., 2019), and GATGNN (Louis et al., 2020) which directly use multi-edge crystal graphs fail short to achieve geometric completeness for crystalline materials.

**Limitations of encoding bond angles using multi-edge crystal graphs**. Noting that further encoding bond angles using line graphs upon multi-edge crystal graphs still fail short to capture periodic patterns, as shown in Fig. 4. Additionally, further encoding bond angles using line graphs upon multi-edge crystal graphs still cannot distinguish crystals shown in Fig. 5 since bond angles are identical for these two crystals. Hence, ALIGNN (Choudhary & DeCost, 2021) and M3GNet (Chen & Ong, 2022) which uses multi-edge crystal graphs with extra bond angles may map crystals with different properties to the same crystal graphs.

**Limitations of Matformer and PotNet**. By directly encoding periodic patterns using self-connecting edges, Matformer (Yan et al., 2022) can distinguish the two crystals shown in Fig. 4. However, Matformer cannot distinguish crystals with the same periodic patterns but different unit cell structures as shown in Fig. 5 due to the fact that the Euclidean distances encoded in the crystal graphs are identical. PotNet (Lin et al., 2023) encodes periodic patterns by using infinite summations of two-body atomic potentials. However, PotNet only considers two-body interactions and using summation operation with errors, resulting in the inability to achieve geometric completeness.

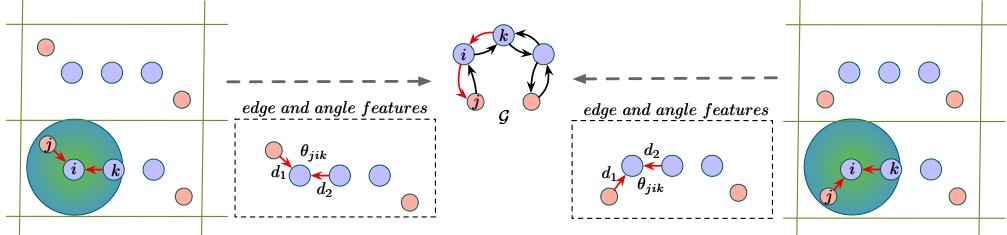

Figure 5: Demonstration that multi-edge crystal graphs fail short to distinguish crystalline materials with different unit cell structures. We use $i, j, k$ to represent three atoms in the unit cell structures of these two crystals, $d_1, d_2$ to represent the encoded bond lengths in the graph, and $\theta_{jik}$ to represent the encoded bond angle between edge $ji$ and $ik$. These two crystals are different but have the identical multi-edge crystal graphs with identical Euclidean distances and bond angles encoded. The number of blue atoms in the middle part is three, and can increase to any arbitrary number to obtain the same conclusion.

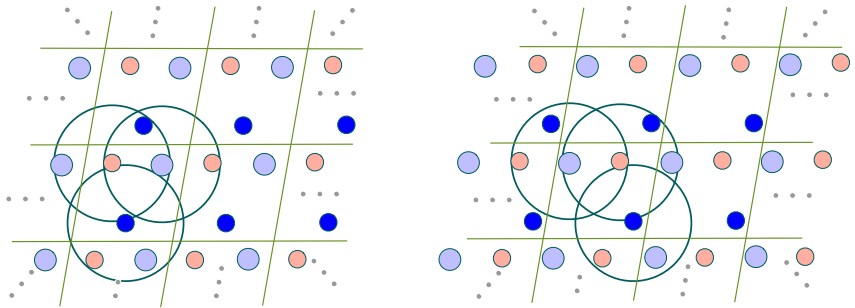

Figure 6: Demonstration that directly using molecule graphs will break periodic invariance. We use green circles to denote the local radius region for every atom in the unit cell. The two crystals shown above are identical and obtained by using periodic transformation (shifting the periodic boundaries). Molecule graphs treat every atom within the radius as a single node in the constructed graph, and will include two blue nodes in total for the left crystal but include three blue nodes in total for the right crystal. This is because molecule graphs break periodic invariance for crystals.

### A.1.2 LIMITATIONS OF PREVIOUS COMPLETE MOLECULE GRAPH NEURAL NETWORKS

To achieve completeness for molecules with finite number of atoms, the molecule graphs used are radius graphs and fully-connected graphs.

**Limitations of methods using radius graphs**. SphereNet (Liu et al., 2022) and ComENet (Wang et al., 2022) use radius graphs, which fail short to capture periodic patterns similar to the multi-edge crystal graphs as shown in Fig. 4. Additionally, molecule graphs break periodic invariance for crystal structures and will generate graphs with different number of nodes for the same crystal as shown in Fig. 6.

**Limitations of methods using fully-connected graphs**. Another approach to achieve geometric completeness for molecules will be constructing fully-connected graphs. However, **crystalline materials are periodic structures**, which makes constructing fully-connected graphs infeasible. Additionally, constructing fully-connected graphs for super cell structures will break periodic invariance as shown in Fig. 7.

### A.1.3 LIMITATIONS OF PREVIOUS COMPLETE CRYSTAL MATRIX FORM REPRESENTATIONS

There are two recent works, AMD-ML (Ropers et al., 2021) and PDD-ML (Balasingham et al., 2022) that use previously established powerful matrix form crystal representations, AMD and PDD.

**Limitations of AMD-ML**. Although AMD-ML uses complete crystal matrix representation AMD, it converts the matrix form representation to a single invariant vector using summation as the input of machine learning algorithms. By doing this transformation, the vector itself cannot fully recover the

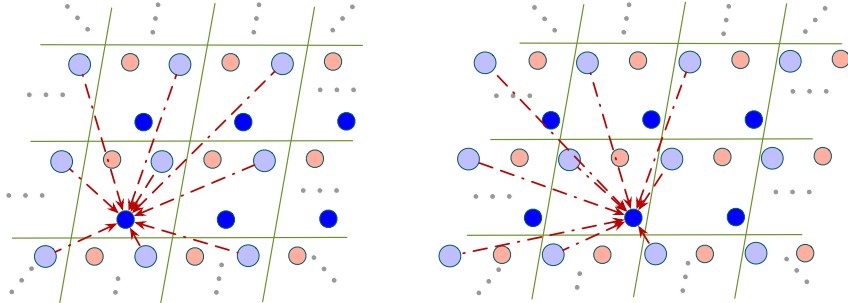

Figure 7: Demonstration that constructing fully-connected graphs for super cell crystal (expanding unit cell structures using surrounding unit cells) structures breaks periodic invariance (cannot remain the same when the periodic boundaries are shifted).

crystal structure and loses some structural information. Additionally, AMD-ML does not consider atom-type information of crystals and can only be used for stable crystal structures.

**Limitations of PDD-ML**. PDD-ML is more powerful than AMD-ML by converting the PDD matrix into crystal graphs like CGCNN-graph. By doing so, PDD-ML considers atom-type information. However, to achieve completeness for the PDD matrix, a large enough $k$ needs to be predetermined and fixed before training for any future test samples, where $k$ needs to be larger than the maximum number of atoms in the cell for all test crystals. This means that the PDD-ML graph needs to have $k$ larger than 200 for The Materials Project, and larger than 700 for the T2 dataset that they tested on. Due to this practical concern, PDD-ML only uses $k_{max} = 25$, which is far from completeness for their experiments on the Materials Project and T2 dataset. Additionally, the geometric information considered in PDD-ML for a fixed number of neighbors is the same as CGCNN with the same $k$ as mentioned by the authors and verified by experiments in their paper. As demonstrated by extensive experimental results, the proposed Comformer is significantly more effective than CGCNN in all tasks. Additionally, by using $k = 25$ nearest neighbors, our proposed graph representations achieve completeness for the T2 dataset as shown in Tab. 8.

## A.2 PROOFS THAT THE PROPOSED CRYSTAL GRAPH REPRESENTATIONS AND COMFORMER SATISFY CRYSTAL PASSIVE SYMMETRIES

In this section, we prove that the proposed invariant crystal graph representation is SE(3) invariant, and the proposed equivariant crystal graph representation is SO(3) equivariant, and both of them are periodic invariant. We use the same notations following the main paper as follows.

Crystal structures are represented by $\mathbf{M} = (\mathbf{A}, \mathbf{P}, \mathbf{L})$, where $\mathbf{A} = [\boldsymbol{a}_1, \boldsymbol{a}_2, \cdots, \boldsymbol{a}_n] \in \mathbb{R}^{d_a \times n}$ contains the $d_a$-dimensional feature vectors for $n$ atoms in the unit cell, $\mathbf{P} = [\boldsymbol{p}_1, \boldsymbol{p}_2, \cdots, \boldsymbol{p}_n] \in \mathbb{R}^{3 \times n}$ contains the 3D Euclidean positions of $n$ atoms in the unit cell, and $\mathbf{L} = [\boldsymbol{\ell}_1, \boldsymbol{\ell}_2, \boldsymbol{\ell}_3] \in \mathbb{R}^{3 \times 3}$ describes the repeating patterns of the unit cell structure in 3D space. Different from molecules and proteins, crystal structures naturally repeat infinitely in space. Formally, the infinite crystal structure can be described as

$$
\begin{aligned}
\hat{\mathbf{P}} &= \{\hat{\boldsymbol{p}}_i | \hat{\boldsymbol{p}}_i = \boldsymbol{p}_i + k_1 \boldsymbol{\ell}_1 + k_2 \boldsymbol{\ell}_2 + k_3 \boldsymbol{\ell}_3, \ k_1, k_2, k_3 \in \mathbb{Z}, i \in \mathbb{Z}, 1 \le i \le n\}, \\
\hat{\mathbf{A}} &= \{\hat{\boldsymbol{a}}_i | \hat{\boldsymbol{a}}_i = \boldsymbol{a}_i, i \in \mathbb{Z}, 1 \le i \le n\},
\end{aligned}
\tag{8}
$$

where the set $\hat{\mathbf{P}}$ represents all 3D positions for every atom $i$ in the unit cell structure, and the set $\hat{\mathbf{A}}$ represents the corresponding atom feature vector for every atom.

### A.2.1 SE(3) INVARIANT CRYSTAL GRAPH REPRESENTATION

As defined in the main paper, a crystal graph representation is SE(3) invariant if, for any rotation transformation $\mathbf{R} \in \mathbb{R}^{3 \times 3}, |\mathbf{R}| = 1$ and translation transformation $b \in \mathbb{R}^3$, the crystal graph remains the same, $f(\mathbf{A}, \mathbf{P}, \mathbf{L}) = f(\mathbf{A}, \mathbf{RP} + b, \mathbf{RL})$. In the following, we prove the SE(3) invariance and periodic invariance step by step following the invariant crystal graph construction process.

First, we construct the lattice representation $\mathbf{e_{ii_1}}, \mathbf{e_{ii_2}}, \mathbf{e_{ii_3}}$ for every node, by using the process demonstrated in Sec.3.3. The selection of $\mathbf{e_{ii_1}}, \mathbf{e_{ii_2}}, \mathbf{e_{ii_3}}$ is purely based on three components, including Euclidean distances, bond angles, and whether $\mathbf{e_{ii_1}}, \mathbf{e_{ii_2}}, \mathbf{e_{ii_3}}$ form the right-hand system. And these three components are naturally SE(3) invariant which remain unchanged under any rotation transformation $\mathbf{R} \in \mathbb{R}^{3 \times 3}, |\mathbf{R}| = 1$ and translation transformation $b \in \mathbb{R}^3$. Additionally, these three components will remain the same for a given node $i$ and all its duplicates whatever unit cell structure you choose to represent the infinite crystal structure. Therefore, the choice of duplicates $i_1, i_2, i_3$ in the constructed lattice representation is SE(3) invariant and periodic invariant. Additionally, due to the fact that reflection transformations will change whether $\mathbf{e_{ii_1}}, \mathbf{e_{ii_2}}, \mathbf{e_{ii_3}}$ form the right-hand system, our proposed invariant crystal graph representation can distinguish chiral crystal structures.

Next, we use the minimum unit cell structure with $n$ atoms and construct a crystal graph with $n$ nodes. This step is already handled by the JARVIS, MP, and MatBench datasets. Due to the fact that all the minimum unit cell structures for a given crystal share the same number of atoms and corresponding atom features, this step is SE(3) invariant and periodic invariant.

Then, we build edges from neighboring nodes to every node $i$. Specifically, an edge will be built from node $j$ to node $i$ when the Euclidean distance $||\mathbf{e_{j'i}}||_2$ between a duplicate of $j$ and $i$ satisfies $||\mathbf{e_{j'i}}||_2 = ||\mathbf{p}_j + k_1' \boldsymbol{\ell}_1 + k_2' \boldsymbol{\ell}_2 + k_3' \boldsymbol{\ell}_3 - \mathbf{p}_i||_2 \leq r$, where $r \in \mathbb{R}$ is the cutoff radius. Duplicates $i_1, i_2$, and $i_3$ are included as neighbors of $i$ to encode periodic patterns. Due to the fact that the Euclidean distance $||\mathbf{e_{j'i}}||_2$ between a duplicate of $j$ and $i$ remains unchanged under SE(3) transformations and across different unit cell structure representations, and the choice of duplicates $i_1, i_2, i_3$ in the constructed lattice representation is SE(3) invariant and periodic invariant as shown in the first step, the neighborhood of node $i$ is SE(3) invariant and periodic invariant.

In the end, we construct edge features for every edge. In the proposed invariant crystal graph, each edge has a corresponding feature vector $\{||\mathbf{e_{j'i}}||_2, \theta_{j'i,ii_1}, \theta_{j'i,ii_2}, \theta_{j'i,ii_3}\}$, where $\theta_{j'i,ii_1}, \theta_{j'i,ii_2}$, and $\theta_{j'i,ii_3}$ denoting the angles between $\mathbf{e_{j'i}}$ and $\{\mathbf{e_{ii_1}}, \mathbf{e_{ii_2}}, \mathbf{e_{ii_3}}\}$, respectively. Due to the fact that $\mathbf{e_{j'i}}$ and $\{\mathbf{e_{ii_1}}, \mathbf{e_{ii_2}}, \mathbf{e_{ii_3}}\}$ rotate accordingly to arbitrary rotation matrix $\mathbf{R} \in \mathbb{R}^{3 \times 3}, |\mathbf{R}| = 1$, the relative bond angles $\theta_{j'i,ii_1}, \theta_{j'i,ii_2}$, and $\theta_{j'i,ii_3}$ remain unchanged under SE(3) transformations and across different unit cell structure representations. Thus, the construct edge features for every edge are SE(3) invariant and periodic invariant.

By combining these four steps in the invariant crystal graph construction process, we complete the proof that the proposed invariant crystal graph representation is SE(3) invariant and periodic invariant.

### A.2.2 SO(3) EQUIVARIANT CRYSTAL GRAPH REPRESENTATION

As defined in the main paper, a crystal graph representation is SO(3) equivariant if, for any rotation transformation $\mathbf{R} \in \mathbb{R}^{3 \times 3}, |\mathbf{R}| = 1$ and translation transformation $b \in \mathbb{R}^3$, the crystal graph rotates accordingly, $\mathbf{R}f(\mathbf{A}, \mathbf{P}, \mathbf{L}) = f(\mathbf{A}, \mathbf{RP} + b, \mathbf{RL})$. In the following, we prove the SO(3) equivariance and periodic invariance step by step following the equivariant crystal graph construction process.

For the lattice representation construction and the node and edge construction, it is the same as the invariant crystal graph. Thus, the choice of duplicates $i_1, i_2, i_3$ in the constructed lattice representation and the node and edge construction is SE(3) invariant and periodic invariant and can distinguish chiral crystal structures.

Then, for the edge features, $||\mathbf{e_{ji}}||_2$ is naturally SE(3) invariant and periodic invariant, while $\mathbf{e_{ji}}$ is SO(3) equivariant and periodic invariant.

Overall, the whole equivariant crystal graph representation is SO(3) equivariant and periodic invariant, with SE(3) invariant edge feature $||\mathbf{e_{ji}}||_2$ and SO(3) equivariant edge feature $\mathbf{e_{ji}}$.

**Relationship between unit cell SE(3) invariant and SO(3) equivariant functions**. Here we show the relationship between unit cell SE(3) invariant and SO(3) equivariant functions. For any SO(3) equivariant function $f$, it satisfies $\mathbf{R}f(\mathbf{A}, \mathbf{P}, \mathbf{L}) = f(\mathbf{A}, \mathbf{RP} + b, \mathbf{RL})$. Then, it can be seen that

$g = ||f||_2$ is a corresponding unit cell SE(3) invariant function. The proof is as follows,

$$g(\mathbf{A}, \mathbf{RP} + b, \mathbf{RL}) = ||f(\mathbf{A}, \mathbf{RP} + b, \mathbf{RL})||_2 \tag{9}$$
$$= ||\mathbf{R}f(\mathbf{A}, \mathbf{P}, \mathbf{L})||_2 \tag{10}$$
$$= ||f(\mathbf{A}, \mathbf{P}, \mathbf{L})||_2 \tag{11}$$
$$= g(\mathbf{A}, \mathbf{P}, \mathbf{L}), \tag{12}$$

Then the proof that $g = ||f||_2$ is a unit cell SE(3) invariant function is complete.

### A.2.3 SE(3) INVARIANCE OF COMFORMER

**iComFormer**. The usage of SE(3) invariant crystal graph representations make the iComFormer SE(3) invariant. Thus, further proof is not needed.

**eComFormer**. The major difference between eComFormer and iComFormer is the usage of vector edge feature $\mathbf{e_{ji}}$ in node-wise equivariant updating layer. Due to the equivariant property of tensor product operations of $\mathbf{TP}_1, \mathbf{TP}_2$, and invariant property of tensor product operation $\mathbf{TP}_0$, after two tensor product layers in node-wise equivariant updating layer, the updated node feature $\boldsymbol{f}_i^{l*}$ is SE(3) invariant.

### A.3 PROOFS OF BASE CASE AND INDUCTIVE STEP IN PROPOSITION 1

**Definition 5** (Isometric crystal structures). *Two crystal structures* $\mathbf{M} = (\mathbf{A}, \mathbf{P}, \mathbf{L})$ *and* $\mathbf{M}' = (\mathbf{A}', \mathbf{P}', \mathbf{L}')$ *are isometric if* $\exists\, \mathbf{R} \in \mathbb{R}^{3 \times 3}, |\mathbf{R}| = 1$ *and* $b \in \mathbb{R}^3$, *such that*

$$\hat{\mathbf{P}}' = \mathbf{R} \bullet \hat{\mathbf{P}} + b = \{\mathbf{R}\hat{\boldsymbol{p}}_i + b | \hat{\boldsymbol{p}}_i = \boldsymbol{p}_i + k_1 \boldsymbol{\ell}_1 + k_2 \boldsymbol{\ell}_2 + k_3 \boldsymbol{\ell}_3,\ k_1, k_2, k_3 \in \mathbb{Z}, i \in \mathbb{Z}, 1 \le i \le n\},$$
$$\hat{\mathbf{A}}' = \hat{\mathbf{A}} = \{\hat{\boldsymbol{a}}_i | \hat{\boldsymbol{a}}_i = \boldsymbol{a}_i, i \in \mathbb{Z}, 1 \le i \le n\},$$

*where* $\hat{\mathbf{P}}, \hat{\mathbf{P}}'$ *are the infinite set representing all 3D positions for every atom in the two crystal structures and* $\hat{\mathbf{A}}, \hat{\mathbf{A}}'$ *are the corresponding atom types as described in Sec. 2.1, and* $\mathbf{R} \bullet \hat{\mathbf{P}} + b$ *describes applying rotation transformation* $\mathbf{R}$ *and translation* $b$ *to every item in the infinite set* $\hat{\mathbf{P}}$.

### A.3.1 PROOFS OF BASE CASE

*Proof.* We employ contradiction to prove it. We first assume there are two different infinite crystal structures with a single atom in the unit cells ($n = 1$) that have the same SO(3) equivariant crystal graph or SE(3) invariant crystal graph, and then demonstrate this assumption leads to contradiction.

(1) Proof for SO(3) equivariant crystal graph. Following the SO(3) equivariant crystal graph construction process in Sec. 3.3, we have obtained lattice representation $\{\mathbf{e_{ii_1}}, \mathbf{e_{ii_2}}, \mathbf{e_{ii_3}}\}$ where $i$ represents the single atom in the unit cells, and $i_1, i_2, i_3$ are three nearest duplicates of node $i$, satisfying right-hand system. Thus, $\mathbf{e_{ii_1}}, \mathbf{e_{ii_2}}, \mathbf{e_{ii_3}}$ form a minimum lattice structure for these two crystals. Then, we can have

$$\hat{\mathbf{P}}_1 = \{\hat{\boldsymbol{p}}_i | \hat{\boldsymbol{p}}_i = \boldsymbol{p}_i + k_1 \mathbf{e_{ii_1}} + k_2 \mathbf{e_{ii_1}} + k_3 \mathbf{e_{ii_1}},\ k_1, k_2, k_3 \in \mathbb{Z}\}, \tag{13}$$
$$\hat{\mathbf{P}}_2 = \{\hat{\boldsymbol{p}}_i{}' | \hat{\boldsymbol{p}}_i{}' = \boldsymbol{p}_i' + k_1 \mathbf{e_{ii_1}} + k_2 \mathbf{e_{ii_1}} + k_3 \mathbf{e_{ii_1}},\ k_1, k_2, k_3 \in \mathbb{Z}\}, \tag{14}$$

where $\hat{\mathbf{P}}_1$ and $\hat{\mathbf{P}}_2$ are the infinite set for atom positions of these two different crystal structures. By using $b = \boldsymbol{p}_i' - \boldsymbol{p}_i$, we can have $\hat{\mathbf{P}}_2 = \hat{\mathbf{P}}_1 + b$. Then, according to Def. 5, it can be seen that these two previously assumed different crystal structures are identical, and it contradicts the assumption.

(2) Proof for SE(3) invariant crystal graph. Following the SE(3) invariant crystal graph construction process in Sec. 3.2, we have obtained lattice representation $\{\mathbf{e_{ii_1}}, \mathbf{e_{ii_2}}, \mathbf{e_{ii_3}}\}$ where $i$ represents the single atom in the unit cells, and $i_1, i_2, i_3$ are three nearest duplicates of node $i$, satisfying right-handed system. Thus, $\mathbf{e_{ii_1}}, \mathbf{e_{ii_2}}, \mathbf{e_{ii_3}}$ form a minimum lattice structure for these two crystals. And duplicates $i_1, i_2$, and $i_3$ are encoded as neighbors of node $i$. Then, we have obtained $\{||\mathbf{e_{i_1 i}}||_2, \theta_{i_1 i, i i_1}, \theta_{i_1 i, i i_2}, \theta_{i_1 i, i i_3}\}$ as the edge feature from $i_1$ to $i$, $\{||\mathbf{e_{i_2 i}}||_2, \theta_{i_2 i, i i_1}, \theta_{i_2 i, i i_2}, \theta_{i_2 i, i i_3}\}$ as the edge feature from $i_2$ to $i$, and $\{||\mathbf{e_{i_3 i}}||_2, \theta_{i_3 i, i i_1}, \theta_{i_3 i, i i_2}, \theta_{i_3 i, i i_3}\}$ as the edge feature from $i_3$ to $i$. Since the SE(3) invariant crystal graph is identical for these two different crystals, these angles are

all identical to each other. Based on the Proof for SO(3) equivariant crystal graph above, we only need to prove the minimum lattice structures of these two crystals are identical, to contradict the assumption that these two crystals are different. Based on the edge features from node $i_1, i_2, i_3$ to $i$, the minimum lattice structures of these two crystals can be recovered as follows.

For the first lattice vector $\mathbf{e_{ii_1}}$ and $\mathbf{e'_{ii_1}}$ for these two crystals, we have

$$\mathbf{e_{ii_1}} = ||\mathbf{e_{i_1i}}||_2\mathbf{x} + 0\mathbf{y} + 0\mathbf{z}, \tag{15}$$

$$\mathbf{e'_{ii_1}} = ||\mathbf{e_{i_1i}}||_2\mathbf{x} + 0\mathbf{y} + 0\mathbf{z}, \tag{16}$$

where $\mathbf{x}, \mathbf{y}, \mathbf{z}$ are unit vectors in a right-hand coordinate system, which means $||\mathbf{x}||_2 = 1, ||\mathbf{y}||_2 = 1, ||\mathbf{z}||_2 = 1$ and $\mathbf{x} \cdot \mathbf{y} = 0, \mathbf{x} \cdot \mathbf{z} = 0, \mathbf{y} \cdot \mathbf{z} = 0$, where $\cdot$ denotes dot product. According to Def. 5, for this single atom case, all other choices of coordinate system can be converted to the above representations by applying a rotation matrix $\mathbf{R} \in \mathbb{R}^{3\times3}, |\mathbf{R}| = 1$.

For the second lattice vector $\mathbf{e_{ii_2}}$ and $\mathbf{e'_{ii_2}}$ for these two crystals, we have

$$\mathbf{e_{ii_2}} = ||\mathbf{e_{i_2i}}||_2 cos(\theta_{i_2i,ii_1})\mathbf{x} + ||\mathbf{e_{i_2i}}||_2 sin(\theta_{i_2i,ii_1})\mathbf{y} + 0\mathbf{z}, \tag{17}$$

$$\mathbf{e'_{ii_2}} = ||\mathbf{e_{i_2i}}||_2 cos(\theta_{i_2i,ii_1})\mathbf{x} + ||\mathbf{e_{i_2i}}||_2 sin(\theta_{i_2i,ii_1})\mathbf{y} + 0\mathbf{z}, \tag{18}$$

Again, according to Def. 5, all other choices to represent the second lattice vector can be converted to the above representations by applying a rotation matrix $\mathbf{R} \in \mathbb{R}^{3\times3}, |\mathbf{R}| = 1$.

Then, to prove that these two crystals share the same minimum lattice structure, we only need to prove the third lattice vector is identical. For the third lattice vectors $\mathbf{e_{ii_3}}$ and $\mathbf{e'_{ii_3}}$ for these two crystals, we have

$$\mathbf{e_{ii_3}} \cdot \mathbf{e_{ii_3}} = \mathbf{e'_{ii_3}} \cdot \mathbf{e_{ii_3}} = ||\mathbf{e_{i_3i}}||_2^2, \tag{19}$$

$$\mathbf{e_{ii_3}} \cdot \mathbf{e_{ii_2}} = \mathbf{e'_{ii_3}} \cdot \mathbf{e_{ii_2}} = ||\mathbf{e_{i_3i}}||_2||\mathbf{e_{i_2i}}||_2 cos(\theta_{i_3i,ii_2}), \tag{20}$$

$$\mathbf{e_{ii_3}} \cdot \mathbf{e_{ii_1}} = \mathbf{e'_{ii_3}} \cdot \mathbf{e_{ii_1}} = ||\mathbf{e_{i_3i}}||_2||\mathbf{e_{i_1i}}||_2 cos(\theta_{i_3i,ii_1}), \tag{21}$$

Then we can have

$$||\mathbf{e_{ii_3}}||_2 = ||\mathbf{e'_{ii_3}}||_2 = ||\mathbf{e_{i_3i}}||_2, \tag{22}$$

$$(\mathbf{e_{ii_3}} - \mathbf{e'_{ii_3}}) \cdot \mathbf{e_{ii_2}} = 0, \tag{23}$$

$$(\mathbf{e_{ii_3}} - \mathbf{e'_{ii_3}}) \cdot \mathbf{e_{ii_1}} = 0, \tag{24}$$

which means $(\mathbf{e_{ii_3}} - \mathbf{e'_{ii_3}}) = \mathbf{0}$ (identical), or $(\mathbf{e_{ii_3}} - \mathbf{e'_{ii_3}}) = c\mathbf{e_{ii_1}} \times \mathbf{e_{ii_2}}$, where $c \in \mathbb{R}$ and $c \neq 0$.

Additionally, due to the right hand system constraint during the graph construction process, we have

$$\mathbf{e_{ii_1}} \times \mathbf{e_{ii_2}} \cdot \mathbf{e_{ii_3}} > 0, \tag{25}$$

$$\mathbf{e_{ii_1}} \times \mathbf{e_{ii_2}} \cdot \mathbf{e'_{ii_3}} > 0. \tag{26}$$

Then we have

$$\mathbf{e_{ii_1}} \times \mathbf{e_{ii_2}} \cdot \mathbf{e_{ii_3}} = \frac{1}{c}(\mathbf{e_{ii_3}} - \mathbf{e'_{ii_3}}) \cdot \mathbf{e_{ii_3}} = \frac{1}{c}(||\mathbf{e_{ii_3}}||_2^2 - \mathbf{e'_{ii_3}} \cdot \mathbf{e_{ii_3}}) > 0, \tag{27}$$

$$\mathbf{e_{ii_1}} \times \mathbf{e_{ii_2}} \cdot \mathbf{e'_{ii_3}} = \frac{1}{c}(\mathbf{e_{ii_3}} - \mathbf{e'_{ii_3}}) \cdot \mathbf{e'_{ii_3}} = \frac{1}{c}(\mathbf{e_{ii_3}} \cdot \mathbf{e'_{ii_3}} - ||\mathbf{e'_{ii_3}}||_2^2) > 0, \tag{28}$$

$$\tag{29}$$

we then can derive that

$$||\mathbf{e_{ii_3}}||_2^2 > \mathbf{e'_{ii_3}} \cdot \mathbf{e_{ii_3}} = \mathbf{e_{ii_3}} \cdot \mathbf{e'_{ii_3}} > ||\mathbf{e'_{ii_3}}||_2^2 \tag{30}$$

which contradicts that $||\mathbf{e_{ii_3}}||_2^2 = ||\mathbf{e'_{ii_3}}||_2^2$. Thus, $(\mathbf{e_{ii_3}} - \mathbf{e'_{ii_3}}) = \mathbf{0}$, and these two crystals have the identical minimum lattice structure. And this contradicts that these two crystals are different. Thus, the proof for SE(3) invariant crystal graph of Base Case is done.

$\square$

A.3.2  PROOFS OF INDUCTIVE STEP

Here we prove the relative position of newly added node $j$ is uniquely determined by the proposed SO(3) equivariant crystal graph and SE(3) invariant crystal graph.

*Proof.* We employ contradiction to prove it. We first assume there are two different relative positions of $j, j'$ that have the same SO(3) equivariant crystal graph or SE(3) invariant crystal graph, and then demonstrate this assumption leads to contradiction.

(1) Proof for SO(3) equivariant crystal graph. Following the SO(3) equivariant crystal graph construction process in Sec. 3.3, for any node $i$ in the $m$ nodes already determined and $i \in \mathcal{N}_j$, we have edge feature $\mathbf{e_{ij}}$. And the relative position of node $j$ and $j'$ according to node $i$ are $\boldsymbol{p}_{ji} = \mathbf{e_{ij}}$ and $\boldsymbol{p}_{j'i} = \mathbf{e_{ij}}$, which contradicts the assumption. The proof is finished.

(2) Proof for SE(3) invariant crystal graph. Following the SE(3) invariant crystal graph construction process in Sec. 3.2, for any node $i$ in the $m$ nodes already determined and $i \in \mathcal{N}_j$, we have edge feature $\{||\mathbf{e_{ji}}||_2, \theta_{ji,ii_1}, \theta_{ji,ii_2}, \theta_{ji,ii_3}\}$. Then, for the relative positions $\boldsymbol{p}_{ji}$ and $\boldsymbol{p}_{j'i}$ of $j$ and $j'$ according to $i$, we have

$$\boldsymbol{p}_{ji} \cdot \mathbf{e_{ii_1}} = ||\mathbf{e_{ji}}||_2||\mathbf{e_{ii_1}}||_2 cos(\theta_{ji,ii_1}) = \boldsymbol{p}_{j'i} \cdot \mathbf{e_{ii_1}}, \tag{31}$$

$$\boldsymbol{p}_{ji} \cdot \mathbf{e_{ii_2}} = ||\mathbf{e_{ji}}||_2||\mathbf{e_{ii_2}}||_2 cos(\theta_{ji,ii_2}) = \boldsymbol{p}_{j'i} \cdot \mathbf{e_{ii_2}}, \tag{32}$$

$$\boldsymbol{p}_{ji} \cdot \mathbf{e_{ii_3}} = ||\mathbf{e_{ji}}||_2||\mathbf{e_{ii_3}}||_2 cos(\theta_{ji,ii_3}) = \boldsymbol{p}_{j'i} \cdot \mathbf{e_{ii_3}}, \tag{33}$$

Then we can have

$$(\boldsymbol{p}_{ji} - \boldsymbol{p}_{j'i}) \cdot \mathbf{e_{ii_1}} = 0, \tag{34}$$

$$(\boldsymbol{p}_{ji} - \boldsymbol{p}_{j'i}) \cdot \mathbf{e_{ii_2}} = 0, \tag{35}$$

$$(\boldsymbol{p}_{ji} - \boldsymbol{p}_{j'i}) \cdot \mathbf{e_{ii_3}} = 0, \tag{36}$$

and because $\mathbf{e_{ii_1}}, \mathbf{e_{ii_1}}, \mathbf{e_{ii_3}}$ form a set of basis for 3D coordinate system, we have $(\boldsymbol{p}_{ji} - \boldsymbol{p}_{j'i}) = \mathbf{0}$, which contradicts the assumption. The proof is complete. □

A.4  VERIFICATION OF GEOMETRIC COMPLETENESS

Geometrically complete crystal graphs can fully recover the input infinite crystal structures. We further verify the geometric completeness of our proposed SE(3) invariant crystal graph and SO(3) equivariant crystal graph by reconstructing the crystal structures from graphs and then compare the reconstructed crystal structures with the original ones.

The reconstruction of infinite crystal structure using the proposed invariant crystal graph is decomposed into four steps, including (1) the reconstruction of lattice representation for each node, (2) the reconstruction of neighboring node positions for a given node, (3) the reconstruction of unit cell structure, and (4) the reconstruction of infinite crystal structure, as shown in Fig. 8. For step one, the information used includes node feature $\boldsymbol{a}_i$, edge features $||\mathbf{e_{ii_1}}||_2, ||\mathbf{e_{ii_2}}||_2, ||\mathbf{e_{ii_3}}||_2$, and $\theta_{i_2i,ii_1}$, $\theta_{i_3i,ii_1}, \theta_{i_3i,ii_2}$, and the right-hand system. For step two, the information used includes node feature $\boldsymbol{a}_j$, edge features $||\mathbf{e_{ji}}||_2$, and $\theta_{ji,ii_1}, \theta_{ji,ii_2}, \theta_{ji,ii_3}$. For step three, the information used is the fact that lattice representation is shared and aligned perfectly for a given crystal across all atoms. For step four, the information used is the constructed lattice representation in step one.

The reconstruction of infinite crystal structure using the proposed equivariant crystal graph can be done similarly. Specifically, in step one, the lattice representation for node $i$ can be directly reconstructed by using $\mathbf{e_{ii_1}}, \mathbf{e_{ii_2}}, \mathbf{e_{ii_3}}$. And in step two, the position of the neighboring node $j$ can be reconstructed by using $\mathbf{e_{ji}}$. The step three and four can be done in the same way following the demonstration in Fig. 8.

We randomly use 500 crystal structures from the Materials Project, and convert them to the proposed SE(3) invariant and SO(3) equivariant crystal graphs using radius $r$ determined by the 16-th nearest neighbors for every atom, and then use above reconstruction steps to recover the crystal structures from graphs. We then compare the structure differences between the recovered ones and the input ones using Pymatgen. As shown in Table. 7, the input crystal structures are fully recovered, with limited structure error (less than $10^{-6}$).

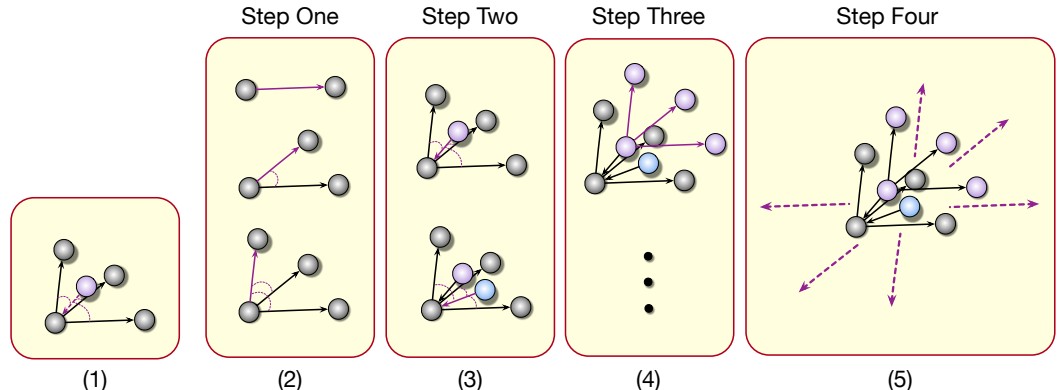

Figure 8: Illustrations of steps reconstructing infinite crystal structure using proposed crystal graph representation. The needed geometric information in each step is marked by purple. (1) The edge information in our crystal graph representation contains the edge length and three angles between three periodic lattice vectors. (2) Construct the three periodic lattice vectors, by only using the edge length to place the first vector, and using the length and relative angle to place the second vector. For the third vector, two relative angles and the bond length are used. Here we choose the right-handed coordinate system for the periodic lattices, thus the orientation is fully determined. (3) By using three angles and the bond length, the position of every neighboring node is determined. (4) Align the lattice vectors for neighboring node since they share the same repeating patterns and repeat step two. (5) Repeat step three until no node is left, and then recover the infinite structure using periodic lattice vectors.

We also extend the completeness verification to the dataset of T2, where a crystal cell can have more than 7 hundred atoms. We use the test set of T2 dataset and convert crystals to the proposed SE(3) invariant and SO(3) equivariant crystal graphs using radius $r$ determined by the 25-th nearest neighbors for every atom, and then use above reconstruction steps to recover the crystal structures from graphs. As shown in Table. 8, the input crystal structures, which can contain more than 700 atoms, are fully recovered with only 25 neighbors, with limited structure error.

Table 7: Verification of geometric completeness of proposed SE(3) invariant and SO(3) equivariant crystal graphs by reconstructing crystal structures from graph representations.

| Structure Differences | Rooted mean square deviation | Mean of largest point-wise distance |
|---|---|---|
| SE(3) invariant | $3.28 * 10^{-7}$ | $4.87 * 10^{-7}$ |
| SO(3) equivariant | $3.03 * 10^{-7}$ | $4.48 * 10^{-7}$ |

Table 8: Verification of geometric completeness of proposed SE(3) invariant and SO(3) equivariant crystal graphs by reconstructing crystal structures from graph representations - T2 test set.

| Structure Differences | Rooted mean square deviation | Mean of largest point-wise distance |
|---|---|---|
| SE(3) invariant | $3.83 * 10^{-7}$ | $2.38 * 10^{-6}$ |
| SO(3) equivariant | $3.19 * 10^{-7}$ | $2.22 * 10^{-6}$ |

A.5    USAGE OF GEOMETRIC INFORMATION IN COMFORMER

In this section, we discuss the usage of geometric information in iComFormer and eComFormer. Specifically, we show that by the proper design of tensor product layers, the output $\boldsymbol{f}_{i,updated}^{l}$ of node-wise equivariant updating layer considers all two-hop bond angle information $\sum_{j \in \mathcal{N}_i} \sum_{m \in \mathcal{N}_j} \mathrm{COS}(\theta_{mj,ji})$ in $\sum_{j \in \mathcal{N}_i} \mathbf{TP}_0(\boldsymbol{f}_{j,1}^{l}, \mathbf{Y}_1(\hat{\mathbf{e}}_{\mathbf{ji}}))$.

**iComFormer**. To begin with, the geometrically complete SE(3) invariant crystal graphs are used as the input of iComFormer. For the iComFormer, the Euclidean distance $||\mathbf{e_{ji}}||_2$ is considered in the node-wise and edge-wise transformer layer, and three bond angles $\theta_{ji,ii_1}$, $\theta_{ji,ii_2}$, $\theta_{ji,ii_3}$ are considered in the edge-wise transformer layer. Therefore, iComFormer considers expressive geometric information of crystals during message passing.

**eComFormer**. The input of eComFormer is the proposed SO(3) equivariant crystal graph. The key question to answer is, how we consider expressive geometric information in eComFormer layers. The proof that the updated node feature $\boldsymbol{f}_{i,updated}^l$ considers bond angle information $\sum_{j\in\mathcal{N}_i}\sum_{m\in\mathcal{N}_j}\mathrm{COS}(\theta_{mj,ji})$ is as follows.

First, $\sum_{j\in\mathcal{N}_i}\mathbf{TP}_0(\boldsymbol{f}_{j,1}^l,\mathbf{Y}_1(\hat{\mathbf{e_{ji}}}))$ is a component in $\boldsymbol{f}_{i,updated}^l$. If $\sum_{j\in\mathcal{N}_i}\mathbf{TP}_0(\boldsymbol{f}_{j,1}^l,\mathbf{Y}_1(\hat{\mathbf{e_{ji}}}))$ considers bond angle information $\sum_{j\in\mathcal{N}_i}\sum_{m\in\mathcal{N}_j}\mathrm{COS}(\theta_{mj,ji})$, then the updated invariant node feature considers $\sum_{j\in\mathcal{N}_i}\sum_{m\in\mathcal{N}_j}\mathrm{COS}(\theta_{mj,ji})$.

We have

$$\sum_{j\in\mathcal{N}_i}\mathbf{TP}_0(\boldsymbol{f}_{j,1}^l,\mathbf{Y}_1(\hat{\mathbf{e_{ji}}})) = \sum_{j\in\mathcal{N}_i}\mathbf{TP}_0(\frac{1}{|\mathcal{N}_i|}\sum_{m\in\mathcal{N}_j}\mathbf{TP}_1(\boldsymbol{f}_m^{l'},\mathbf{Y}_1(\hat{\mathbf{e_{mj}}})),\mathbf{Y}_1(\hat{\mathbf{e_{ji}}})) \tag{37}$$

$$=\frac{1}{|\mathcal{N}_i|}\sum_{j\in\mathcal{N}_i}\mathbf{TP}_0(\sum_{m\in\mathcal{N}_j}\mathbf{TP}_1(\boldsymbol{f}_m^{l'},\mathbf{Y}_1(\hat{\mathbf{e_{mj}}})),\mathbf{Y}_1(\hat{\mathbf{e_{ji}}})) \tag{38}$$

$$=\frac{1}{|\mathcal{N}_i|}\sum_{j\in\mathcal{N}_i}\mathbf{TP}_0(\sum_{m\in\mathcal{N}_j}\mathbf{TP}_1(\boldsymbol{f}_m^{l'},\mathbf{Y}_1(\hat{\mathbf{e_{mj}}})),c_1\hat{\mathbf{e_{ji}}}) \tag{39}$$

$$=\frac{1}{|\mathcal{N}_i|}\sum_{j\in\mathcal{N}_i}\mathbf{TP}_0(\sum_{m\in\mathcal{N}_j}v_m\hat{\mathbf{e_{mj}}},c_1\hat{\mathbf{e_{ji}}}), \tag{40}$$

where $\mathbf{TP}_1(\boldsymbol{f}_m^{l'},\mathbf{Y}_1(\hat{\mathbf{e_{mj}}})) = v_m\hat{\mathbf{e_{mj}}}$, and $v_m$ conditioned on the invariant node feature $\boldsymbol{f}_m^{l'}$ of node $m\in\mathcal{N}_j$, and learnable parameters (path weight) in $\mathbf{TP}_1$ . Here for simplicity, we only show the one output channel case of $\mathbf{TP}_1$, and in the implementation, the output can be multi-channel. Given two vector input $\mathbf{v_1}\in\mathbb{R}^3$, $\mathbf{v_2}\in\mathbb{R}^3$, we have

$$\mathbf{TP}_0(\mathbf{v_1},\mathbf{v_2}) =v_{\mathrm{TP}_0}*(\mathbf{v_{1,x}}*\mathbf{v_{2,x}}+\mathbf{v_{1,y}}*\mathbf{v_{2,y}}+\mathbf{v_{1,z}}*\mathbf{v_{2,z}}) \tag{41}$$

$$=v_{\mathrm{TP}_0}*\mathrm{COS}(\theta_{v_1,v_2})*||\mathbf{v_1}||_2*||\mathbf{v_2}||_2, \tag{42}$$

where $v_{\mathrm{TP}_0}$ is a learnable scalar value in $\mathbf{TP}_0$, and COS describes cosine function. And we have $\mathbf{TP}_0(c\mathbf{v_1}+\mathbf{v_3},\mathbf{v_2}) = c\mathbf{TP}_0(\mathbf{v_1},\mathbf{v_2})+\mathbf{TP}_0(\mathbf{v_3},\mathbf{v_2})$ due to the bi-linear property of $\mathbf{TP}_0$. Based on these, we can further have,

$$\sum_{j\in\mathcal{N}_i}\mathbf{TP}_0(\boldsymbol{f}_{j,1}^l,\mathbf{Y}_1(\hat{\mathbf{e_{ji}}})) =\frac{1}{|\mathcal{N}_i|}\sum_{j\in\mathcal{N}_i}\mathbf{TP}_0(\sum_{m\in\mathcal{N}_j}v_m\hat{\mathbf{e_{mj}}},c_1\hat{\mathbf{e_{ji}}}) \tag{43}$$

$$=\frac{1}{|\mathcal{N}_i|}\sum_{j\in\mathcal{N}_i}\sum_{m\in\mathcal{N}_j}v_{\mathrm{TP}_0}*v_m*c_1*\mathrm{COS}(\theta_{mj,ji}) \tag{44}$$

Thus, $\sum_{j\in\mathcal{N}_i}\mathbf{TP}_0(\boldsymbol{f}_{j,1}^l,\mathbf{Y}_1(\hat{\mathbf{e_{ji}}}))$ considers all two hop bond angle information. And therefore it considers all angle information $\theta_{j_1j,ji}$, $\theta_{j_2j,ji}$, $\theta_{j_3j,ji}$ in the invariant crystal graph representation due to $j_1,j_2,j_3\in\mathcal{N}_j$. Beyond this, the node-wise transformer layer captures geometric information of $||\mathbf{e_{ji}}||_2$, which further captures disturbs in pairwise Euclidean distances.

To sum up, the node-wise equivariant updating layer contains bond angle information between $\mathbf{e_{ji}}$ and $\mathbf{e_{kj}}$, where $j\in\mathcal{N}_i$ and $k\in\mathcal{N}_j$, and therefore considers all 2-hop angle information. And the node-wise transformer layer contains edge lengths $||\mathbf{e_{ji}}||_2$. By combining them together, eComFormer considers expressive geometric information. And if lattice representation including $i_1,i_2,i_3$ is not encoded as the neighbor of node $i$, the infinite crystal structure, especially the periodic patterns, cannot be fully captured.

## A.6   MODEL SETTINGS OF COMFORMER

We provide detailed model settings of iComFormer and eComFormer, and demonstrate the hyperparameter settings of them for different datasets and tasks in this section.

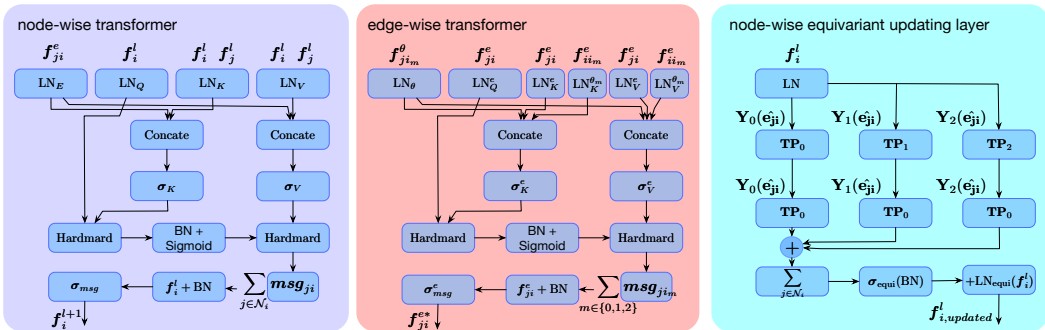

Figure 9: Illustration of the detailed network architectures of the proposed node-wise transformer layer, edge-wise transformer layer, and node-wise equivariant updating layer.

**Detailed network architectures**. We first show the detailed network architectures of the proposed node-wise transformer layer, edge-wise transformer layer, and node-wise equivariant updating layer in Fig. 9. For all the experiments across three crystal datasets, we use two node-wise transformer layers, and one node-wise equivariant updating layer in between to form the eComFormer, and use one node-wise transformer layer, followed by one edge-wise transformer layer, and then $l-1$ node-wise transformer layers to form the $l$ layer iComFormer.

**Settings of embedding layers of ComFormer**. iComFormer and eComFormer both embed node feature $\boldsymbol{a}_1$ (atomic number) to a CGCNN embedding vector of length 92 and then mapped to a 256 dimensional initial node feature $\boldsymbol{f}_i^0$ by a linear transformation. iComFormer and eComFormer both embed invariant edge feature $||\mathbf{e}_{\mathbf{j'i}}||_2$ to $-0.75/||\mathbf{e}_{\mathbf{j'i}}||_2$, and then increase it to 256 dimensional vector by RBF kernels with 256 center value from -4.0 to 0.0, after that, both of them transfer the 256 dimensional vector to initial edge feature $\boldsymbol{f}_{j'i}^e$ by a linear transformation layer and a softplus activation layer. iComFormer embeds the invariant edge feature $\theta_{j'i,ii_1}$, $\theta_{j'i,ii_2}$, and $\theta_{j'i,ii_3}$ to $\boldsymbol{f}_{j'i_1}^\theta$, $\boldsymbol{f}_{j'i_2}^\theta$, and $\boldsymbol{f}_{j'i_3}^\theta$ by taking the consine values and RBF kernels with 256 center values from -1.0 to 1.0, followed by a linear layer and a softplus activation layer. eComFormer embeds the equivariant edge feature $\mathbf{e}_{\mathbf{j'i}}$ to spherical harmonics with rotation order 0, 1, and 2 by using e3nn (Geiger & Smidt, 2022b) library.

**Settings of node-wise transformer layer**. $\mathrm{LN}_Q, \mathrm{LN}_K, \mathrm{LN}_V, \mathrm{LN}_E$ are linear transformation layers that map 256 dimensional input features to 256 dimensional output features. $\boldsymbol{\sigma}_K, \boldsymbol{\sigma}_V$ are the nonlinear transformations for key and value, including one linear layer that maps the concatenated $256*3$ dimensional input features to 256 dimensional output features, one silu activation layer, and one linear layer that maps the 256 dimensional input features to 256 dimensional output features. $\boldsymbol{\sigma}_{msg}$ is the softplus activation layer.

**Settings of edge-wise transformer layer**. Similar to node-wise transformer layer, linear transformations including $\mathrm{LN}_Q^e, \mathrm{LN}_K^e, \mathrm{LN}_V^e, \mathrm{LN}_K^{\theta_m}, \mathrm{LN}_V^{\theta_m}, \mathrm{LN}_\theta$ map 256 dimensional input features to 256 dimensional output features. $\boldsymbol{\sigma}_K^e, \boldsymbol{\sigma}_V^e$ are the nonlinear transformations for key and value, including one linear layer that maps the concatenated $256*3$ dimensional input features to 256 dimensional output features, one silu activation layer, and one linear layer that maps the 256 dimensional input features to 256 dimensional output features. $\boldsymbol{\sigma}_{msg}^e$ is the softplus activation layer.

**Settings of node-wise equivariant updating layer**. We directly use pherical harmonics $\mathbf{Y}_0(\hat{\mathbf{e}_{\mathbf{ji}}}) = c_0$, $\mathbf{Y}_1(\hat{\mathbf{e}_{\mathbf{ji}}}) = c_1 * \hat{\mathbf{e}_{\mathbf{ji}}} \in \mathbb{R}^3$ and $\mathbf{Y}_2(\hat{\mathbf{e}_{\mathbf{ji}}}) \in \mathbb{R}^5$ implemented in e3nn. LN transfers input node feature $\boldsymbol{f}_i^l$ to $\boldsymbol{f}_i^{l'}$. $\mathbf{TP}_0(\boldsymbol{f}_j^{l'}, \mathbf{Y}_0(\hat{\mathbf{e}_{\mathbf{ji}}}))$ is the e3nn tensor product layer with rotation order 0 output features and 128 channels. $\mathbf{TP}_1(\boldsymbol{f}_j^{l'}, \mathbf{Y}_1(\hat{\mathbf{e}_{\mathbf{ji}}}))$ and $\mathbf{TP}_2(\boldsymbol{f}_j^{l'}, \mathbf{Y}_1(\hat{\mathbf{e}_{\mathbf{ji}}}))$ are the e3nn tensor product layer with rotation order 1 and 2 output features and 8 channels. $\mathbf{TP}_0(\boldsymbol{f}_{j,0}^l, \mathbf{Y}_0(\hat{\mathbf{e}_{\mathbf{ji}}})), \mathbf{TP}_0(\boldsymbol{f}_{j,1}^l, \mathbf{Y}_1(\hat{\mathbf{e}_{\mathbf{ji}}})), \mathbf{TP}_0(\boldsymbol{f}_{j,2}^l, \mathbf{Y}_2(\hat{\mathbf{e}_{\mathbf{ji}}}))$ are the e3nn tensor product layer with rotation order 0 output features and 128 channels. $\boldsymbol{\sigma}_{\mathrm{equi}}$ is the nonlinear transformation includes a softplus layer, a linear transformation layer that maps 128 dimensional features to 256 dimensional features, and a softplus layer. And $\mathrm{LN}_{\mathrm{equi}}$ is a linear transformation layer that maps 256 dimensional input features to 256 dimensional output features.

**Importance of higher rotation order features**. As illustrated in Table 9, the inclusion of equivariant updating layer with features $\sum_{j \in \mathcal{N}_i} \mathbf{TP}_0(\boldsymbol{f}_{i,1}^l, \mathbf{Y}_1(\hat{\mathbf{e}}_{\mathbf{ji}}))$ and $\sum_{j \in \mathcal{N}_i} \mathbf{TP}_0(\boldsymbol{f}_{i,2}^l, \mathbf{Y}_2(\hat{\mathbf{e}}_{\mathbf{ji}}))$ with rotation orders one and two consistently enhances the performance of eComFormer from 30.2 to 28.4.

Table 9: Ablation studies. o denotes the max rotation order of features in the equivariant layer.

| Method | node layer | equivariant layer | Test MAE |
|---|---|---|---|
| eComFormer | 3 | 0 | 30.2 |
| eComFormer | 3 | 1 (o = 1) | 29.8 |
| eComFormer | 3 | 1 (o = 2) | **28.4** |

**Graph level prediction**. After the transformer layers, we aggregate the node features in the graph by mean pooling, and then use one linear layer to map the 256 dimensional graph level feature to 256 dimensional output feature, and then use a silu activation layer, and then map the output to a scalar value by a linear transformation layer.

### A.6.1 HYPERPARAMETER SETTINGS OF COMFORMER FOR DIFFERENT TASKS

In this subsection, we share the detailed hyperparameter settings of iComFormer and eComFormer for different tasks in these three crystal datasets. We slightly tuned the parameters of our methods, and higher performances are expected if hyperparameters are further tuned for different tasks.

Table 10: Model settings of iComFormer for JARVIS.

| | Num. node-wise transformer | Num. edge-wise transformer | Learning rate | Num. neighbors |
|---|---|---|---|---|
| formation energy | 4 | 1 | 0.001 | 25 |
| band gap (OPT) | 4 | 1 | 0.0005 | 25 |
| band gap (MBJ) | 4 | 1 | 0.001 | 12 |
| total energy | 4 | 1 | 0.0005 | 25 |
| Ehull | 4 | 1 | 0.001 | 25 |

Table 11: Model settings of eComFormer for JARVIS.

| | Num. node-wise transformer | Num. node-wise equivariant updating | Learning rate | Num. neighbors |
|---|---|---|---|---|
| formation energy | 3 | 1 (max order=2) | 0.001 | 25 |
| band gap (OPT) | 3 | 1 (max order=2) | 0.0005 | 25 |
| band gap (MBJ) | 2 | 1 (max order=2) | 0.001 | 16 |
| total energy | 2 | 1 (max order=2) | 0.001 | 16 |
| Ehull | 3 | 1 (max order=2) | 0.001 | 25 |

Table 12: Model settings of iComFormer for the Materials Project.

| | Num. node-wise transformer | Num. edge-wise transformer | Learning rate | Num. neighbors |
|---|---|---|---|---|
| formation energy | 4 | 1 | 0.0007 | 25 |
| band gap | 4 | 1 | 0.0005 | 25 |
| bulk moduli | 3 | 1 | 0.0001 | 16 |
| shear moduli | 3 | 1 | 0.0001 | 25 |

Table 13: Model settings of eComFormer for the Materials Project.

| | Num. node-wise transformer | Num. node-wise equivariant updating | Learning rate | Num. neighbors |
|---|---|---|---|---|
| formation energy | 3 | 1 (max order=2) | 0.001 | 25 |
| band gap | 2 | 1 (max order=2) | 0.001 | 16 |
| bulk moduli | 2 | 1 (max order=2) | 0.001 | 16 |
| shear moduli | 2 | 1 (max order=2) | 0.001 | 16 |

Table 14: Model settings of iComFormer for MatBench.

| | Num. node-wise transformer | Num. edge-wise transformer | Learning rate | Num. neighbors |
|---|---|---|---|---|
| e_form | 4 | 1 | 0.001 | 25 |
| jdft2d | 4 | 1 | 0.0001 | 16 |

**JARVIS**. We show the model settings of iComFormer and eComFormer in Tables 10 and 11. Specifically, iComFormer is trained using MSE loss with Adam (Kingma & Ba, 2015) optimizer

Table 15: Model settings of eComFormer for MatBench.

| | Num. node-wise transformer | Num. node-wise equivariant updating | Learning rate | Num. neighbors |
|---|---|---|---|---|
| e_form | 2 | 1 (max order=2) | 0.001 | 25 |
| jdft2d | 2 | 1 (max order=2) | 0.0001 | 16 |

for the tasks of MBJ bandgap and Ehull for 500 epochs, and formation energy for 700 epochs, with Onecycle (Smith & Topin, 2018) scheduler and pct_start of 0.3. For OPT bandgap and total energy, iComFormer are trained using L1 loss with Adam (Kingma & Ba, 2015) optimizer and polynomial scheduler with starting learning rate 0.0005 and final learning rate 0.00001 for 800 epochs. eComFormer is trained using MSE loss with Adam (Kingma & Ba, 2015) optimizer for all tasks for 500 epochs except OPT bandgap, with Onecycle (Smith & Topin, 2018) scheduler and pct_start of 0.3. For OPT bandgap, eComFormer is trained using L1 loss with Adam (Kingma & Ba, 2015) optimizer and polynomial scheduler with starting learning rate 0.0005 and final learning rate 0.00001 for 800 epochs. The number of neighbors $k$ indicates the $k$-th nearest distance we use as the radius for node $i$.

**The Materials Project**. We show the model settings of iComFormer and eComFormer in Table 12 and 13. Except bandgap of iComformer, all iComFormer and eComFormer models for the Materials Project are trained for 500 epochs using MSE loss with Adam (Kingma & Ba, 2015) optimizer with Onecycle (Smith & Topin, 2018) scheduler and pct_start of 0.3. For bandgap, iComformer is trained using L1 loss with Adam (Kingma & Ba, 2015) optimizer and polynomial scheduler with starting learning rate 0.0005 and final learning rate 0.00001 for 500 epochs.

**MatBench**. We show the model settings of iComFormer and eComFormer in Table 14 and 15. All iComFormer and eComFormer models for the Materials Project are trained for 500 epochs using MAE loss with Adam (Kingma & Ba, 2015) optimizer. And we use Onecycle (Smith & Topin, 2018) scheduler and pct_start of 0.3.

### A.6.2 EFFICIENCY ANALYSIS WHEN INCREASING $k$

We provide further efficiency analysis for our proposed iComFormer and eComformer when increasing $k$, where $k$ indicates that we use $k$-th smallest distance from neighbors to the center to serve as the radius for node $i$.

We show the running time per epoch on the task of JARVIS formation energy for different ComFormer variants and different k values in Fig. 10. It can be seen that the running time increases linearly for all ComFormer variants when increasing k.

### A.7 DATASET DETAILS

We provide more dataset details for the Materials Project, JARVIS, and MatBench in this section.

**JARVIS**. JARVIS-2021.8.18 is first used by ALIGNN (Choudhary & DeCost, 2021). We follow the experimental settings of Matformer (Yan et al., 2022) and PotNet (Lin et al., 2023) and evaluate our methods on five important crystal property prediction tasks. The training, evaluating, and testing sets for formation energy, total energy, and bandgap(OPT) prediction tasks contain 44578, 5572, and 5572 crystals while contain 44296, 5537, and 5537 crystals for Ehull, and contain 14537, 1817, 1817 for bandgap(MBJ). We follow previous works and use mean absolute error as the evaluation metric. We directly use the benchmark results from Matformer (Yan et al., 2022) and PotNet (Lin et al., 2023). For the efficiency comparisons with previous works, we directly follow the efficiency comparison settings of Matformer (Yan et al., 2022) and use a single RTX A6000 to test the running time per epoch. Among these crystal structures, 18865 crystals are experimentally observed.

**The Materials Project**. The Materials Project-2018.6.1 was first proposed and used by MEG-NET (Chen et al., 2019) collected from The Materials Project (Jain et al., 2013), but the methods are compared by using different random seeds and dataset sizes. To make a fair comparison, Yan et al. (2022) re-train all baseline methods using the same data splits. We follow the experimental settings and data splits of Matformer (Yan et al., 2022) and PotNet (Lin et al., 2023). Specifically, the formation energy and bandgap prediction tasks contain 60000, 5000, and 4239 crystals for training,

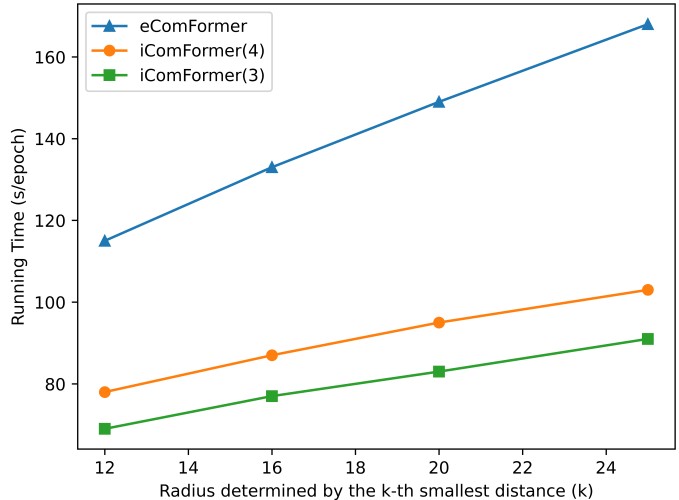

Figure 10: Efficiency analysis for iComFormer and eComformer when increasing $k$, where $k$ indicates the $k$-th smallest distance used as the radius for node $i$. We show the running time per epoch on the task of JARVIS formation energy.

Table 16: MP experimentally observed dataset.

| Method | Number of training crystals | Test MAE |
|---|---|---|
| CGCNN (k=12) | 29342 | 0.051 |
| PDD-best (k=12) | 29342 | 0.047 |
| AMD | 24067 | 0.78 |
| iComformer (k=12) | 24067 | **0.027** |

evaluating, and testing, respectively. And the shear and bulk moduli tasks contain 4664, 393, and 393 crystals for training, evaluating, and testing, respectively. Note that one validation sample in Shear Moduli is removed due to negative GPa indicating the underlying unstable/metastable crystal structure. The evaluation metric used is mean absolute error. We directly use the benchmark results from Matformer (Yan et al., 2022) and PotNet (Lin et al., 2023). Among these crystal structures, 30084 crystals are experimentally observed. We also further conduct experiments on the pure experimentally observed crystal dataset with 30084 crystals, and use random splits 80%, 10%, 10% to form the training, evaluating, and test sets, and compare with CGCNN, PDD, and AMD as shown in Table 16.

**MatBench**. MatBench (Dunn et al., 2020) is a crystal property prediction benchmark, containing tasks with various data scales. To evaluate the robustness of our proposed methods, we conduct experiments on e_form with 132752 crystals and jdft2d with 636 crystals. It is worth noting that jdft2d is a 2D crystal dataset, curated by Matbench. The traditional way (also in Matbench) to represent 2D structure is setting $||l_3||_2 > 10$ Angstroms and not in the same plane of $l_1$, $l_2$. We directly use the official MatBench package and guidelines to run these tasks, and report the benchmark results using metric mean absolute error (MAE) and root mean square error (RMSE) with error bars indicating the standard deviation of five benchmark runs. The benchmark results of our methods will be updated to MatBench in the future. We directly use the benchmark results from MatBench.

## A.8 DISCUSSIONS OF POTENTIAL CORNER CASES

In this section, we discuss the potential corner cases about geometric completeness of the proposed crystal graphs, and potential corner cases due to the usage of the nearest neighbor algorithm, to aid future users when they deploy the proposed method for their own applications.

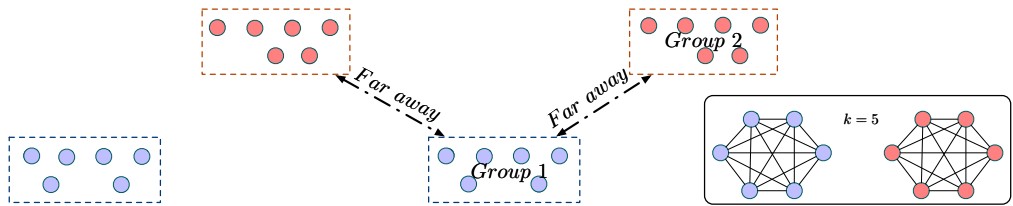

Figure 11: Demonstration of the hypothesis corner case that the constructed graph is not strongly connected. In this hypothesis case, the minimum unit cell structure contains two groups of atoms, marked by blue and pink. These two groups of atoms are far away from each other. We construct crystal graphs using radius $r$ determined by the $k$-th smallest Euclidean distances from neighbors to the center node. For this corner case, if the $k = 5$, there will be no connections between these two groups of nodes, resulting in a not strongly connected graph. A solution to make the graph strongly connected is to increase the value of $k$ and make $k > 5$ (e.g., $k = 6$ will result in a strongly connected graph).

### A.8.1 POTENTIAL CORNER CASES OF GEOMETRIC COMPLETENESS

To achieve geometric completeness for crystalline materials, the only assumption we need is the constructed crystal graph is strongly connected, which means for any node pair $i, j$, where $i, j$ represent the index of atoms in the unit cell, there is a path from $i$ to $j$.

**Hypothesis corner cases**. We then demonstrate the hypothesis cases that the constructed graph is not strongly connected and show how to tackle this issue. As shown in Fig. 11, if there are several groups that are far away from each other and the radius is not large enough, the constructed graph will not be strongly connected. It is worth noting that for a not strongly connected graph, no previous graph representations can achieve completeness, because of the fact that no geometric information between groups is captured.

**Solution**. Although this kind of hypothesis crystalline material may not exist in reality, a solution for this Hypothesis case will be to increase the radius or increase the value of $k$ (when you determine radius $r$ by the $k$-th smallest Euclidean distances from neighbors to the center node).

### A.8.2 POTENTIAL CORNER CASES OF USING NEAREST NEIGHBORS

When a reference node is needed, a widely used E(3) invariant way to choose the reference node is using Euclidean distance and nearest neighbors. Similar to SphereNet (Liu et al., 2022) and ComENet (Wang et al., 2022) which employ the nearest neighbor algorithm to choose reference nodes, our method also uses the Euclidean distance based nearest neighbor algorithm to determine the lattice representations to achieve rotation and translation invariance.

**Potential corner case**. When several periodic duplicates of the atom in a given crystal share the same Euclidean distances, e.g., $||\mathbf{e}_{\mathbf{ii}_1}||_2 = ||\mathbf{e}_{\mathbf{ii}_2}||_2 = ||\mathbf{e}_{\mathbf{ii}_3}||_2$, these duplicates will share the equal possibility to be selected as the reference node.

**Solution and discussion**. This issue occurs for algorithms using nearest neighbors, including SphereNet (Liu et al., 2022), ComENet (Wang et al., 2022), CGCNN (Xie & Grossman, 2018), and GATGNN (Louis et al., 2020). If the nearest neighbor algorithm employed is deterministic, there will be a unique graph for the corner case. If the nearest neighbor algorithm employed is not deterministic, these several different representations of the corner case will occur with equal possibility during the training process (similar to data augmentation).

If the user plans to further restrict the result of the nearest neighbor algorithm, additional information can be used to select from these candidates with the same Euclidean distances. In any case, this issue will not influence the completeness of the constructed representation.

