# OpenReview forum: "Complete and Efficient Graph Transformers for Crystal Material Property Prediction"
_ICLR.cc/2024/Conference — ICLR 2024 poster_

### Official Review · Reviewer_HGjX · 2023-10-30

**Soundness:** 2 fair
**Presentation:** 3 good
**Contribution:** 3 good
**Rating:** 6
**Confidence:** 5

**Summary:**

Constructing graphs that effectively capture the complete geometric information of crystals remains an unsolved and challenging problem. In order to address it, this paper introduced a novel approach that utilizes the periodic patterns of unit cells to establish the lattice-based representation for each atom, enabling efficient and expressive graph representations of crystals.

**Strengths:**

1) The presentation of this paper is good. The author's statement is logical, and the vocabulary used is relatively easy to understand.

2) This paper proposes an interesting problem. A complete crystal representation is important for downstream tasks.

**Weaknesses:**

1) This method is not universal as it cannot handle crystals with lattice containing (0,0,0). The paper mentions that “Third, we select the duplicate with the next smallest distance ||e_ii3||_2, and verify that eii3, eii1 and eii2 are not in the same plane in 3D space, and repeat until eii3 is found”, but if lattice containing (0,0,0), then eii3 cannot be found.

2) The experiment is not comprehensive enough. The most significant contribution of this paper is that it proposes a complete representation, but the final experiment only verifies that it can better predict without verifying the completeness of the representation.

**Questions:**

1) How to deal with the crystal whose lattice contains (0,0,0)? Do the datasets in experiments contain this type of data? In theory, the method proposed by the author cannot handle this situation. Why was it not explained in experiments?

2) What is the role of SE(3) invariant and SO(3) equivariant? In what scenarios should eComFormer be used, and in what scenarios should iComFormer be used?  What are the reasons for iComFormer outperforming eComFormer in most cases?

Suggestions:
If possible, it is recommended that the author add some experiments that demonstrate the completeness of the representation.

---

> ### Author Response · Authors · 2023-11-16
> **Responses from authors**
>
> Thank you for your recognition of the presentation and the problem we focused on.
>
> 1. This method is not universal as it cannot handle crystals with lattice containing (0,0,0). The paper mentions that “Third, we select the duplicate with the next smallest distance ||e_ii3||_2, and verify that eii3, eii1, and eii2 are not in the same plane in 3D space, and repeat until eii3 is found”, but if lattice containing (0,0,0), then eii3 cannot be found. How to deal with the crystal whose lattice contains (0,0,0)? Do the datasets in experiments contain this type of data? In theory, the method proposed by the author cannot handle this situation. Why was it not explained in experiments?
>
> Thank you for this question. For a 3D crystal structure, following the traditional and widely-used definition, if we have a third lattice vector $l_3$ = (0, 0, 0), it means for a given atom i in the cell with position p_i, there will have **infinite number of atoms at position $p_i$**, due to $p_i' = p_i + k_3 \* l_3 = p_i$. Thus, this scenario will not happen in reality.
> If you are talking about purely 2D crystals with a single layer and strictly all z=0, then only the first two lattice vectors are needed to achieve completeness.
>
> For 2D crystals not strictly have all z=0, the traditional way to represent the structure is setting $||l_3||_2 > 10$ Angstroms and not in the same plane of  $l_1$, $l_2$, and the proposed method can naturally handle this case.
>
> It is worth noting that **our experiments include 2D crystals - matbench jdft2d** as shown in the main paper. To make this clear, we revised the paper accordingly and added more details about jdft2d in the experimental section and the appendix.
>
> 2. The experiment is not comprehensive enough. The most significant contribution of this paper is that it proposes a complete representation, but the final experiment only verifies that it can better predict without verifying the completeness of the representation. If possible, it is recommended that the author add some experiments that demonstrate the completeness of the representation.
>
> Thank you for this question. In the original version, we put the completeness verification experiments in Appendix A.4. We do believe in the importance of completeness of our proposed representations, so we follow your suggestion and further conduct completeness verification of our graph representations on a more challenging T2 crystal dataset in which the largest crystal cell contains more than 700 atoms. It can be seen that the proposed graph representations can fully reconstruct the super large crystal structures with only k=25 (where k means k nearest neighbors). Specifically, We convert crystal structures to the proposed SE(3) invariant and SO(3) equivariant crystal graphs and then use reconstruction steps shown in Appendix A.4 to recover the crystal structures from the graphs. We then compare the structure differences between the recovered ones and the input ones using Pymatgen. As shown in the below table, the input crystal structures are fully recovered, with limited structure error (less than $10^{-6}$).
>
> | RMSD | MP Crystals | T2 Crystals [1] |
> | -- | -- | -- |
> |SE(3) invariant| $3.28* 10^{-7}$| $3.83* 10^{-7}$|
> |SO(3) equivariant|$3.03* 10^{-7}$|$3.19* 10^{-7}$|
>
> According to your suggestion, we further **moved the completeness verification experiments from the Appendix to the main paper**, to enhance the clarity.
>
> [1] T2 dataset: Pulido, Angeles, et al. "Functional materials discovery using energy–structure–function maps." Nature 543.7647 (2017): 657-664.
>
> 3. What is the role of SE(3) invariant and SO(3) equivariant? In what scenarios should eComFormer be used, and in what scenarios should iComFormer be used? What are the reasons for iComFormer outperforming eComFormer in most cases?
>
> For the tasks of scalar property prediction, SE(3) and SO(3) can both be used. However, for higher tensor order crystal property predictions, only SO(3) will be suitable. We suggest the usage of iComFormer for scalar property predictions and the usage of eComformer for the development of higher tensor order predictions, which is also one of our future focus.
>
> The reason that iComFormer performed a little better than eComFormer (actually only marginal in many cases, because both of them use complete graphs) is due to the equivariant tensor product layers may require more training samples and are harder to train than invariant features because equivariant features cannot support nonlinear mappings.
>
> We believe your suggestions enhanced the clarity of this paper.
>
> Yours sincerely, authors

---

> ### Comment · Reviewer_HGjX · 2023-11-21
>
> Thank you for the author's response, and most of my doubts and concerns have been addressed.
>
> Here are some of my new questions.
>
> The role of eComformer in predicting the properties of tensor ordered crystals has not been reflected in this work, so it lacks the persuasiveness to consider this as the difference between the two methods. I would like to know the role of eComformer in this work and the differences between these two methods in the tasks involved in this paper.
>
> In addition, I have another question that I would like to discuss with the author. Given that I am not an expert in the field of materials, I am curious to know whether graphene, being a two-dimensional structure in the real world, encounters the issue of the lattice containing (0,0,0).

---

> > ### Author Response · Authors · 2023-11-21
> > **Responses from authors for follow-up questions [Part 1]**
> >
> > Thank you very much for your responses, we are glad that most of your doubts and concerns have been addressed. For your new questions, we provide point-to-point responses as follows.
> >
> > 1. The role of eComformer in predicting the properties of tensor ordered crystals has not been reflected in this work, so it lacks the persuasiveness to consider this as the difference between the two methods. I would like to know the role of eComformer in this work and the differences between these two methods in the tasks involved in this paper.
> >
> > Thank you for this question, the differences between iComFormer and eComFormer are detailed below.
> >
> > First, **invariant iComFormer only considers invariant features and can only be used to predict scalar properties, while eComFormer maintains vector features and can be used to predict higher tensor order properties**. To further demonstrate this, we can use the following example to show that iComFormer cannot predict vector properties. For instance, if the prediction target is $\sum_{j \in N_i} e_{ij} \in R^3$, which is a node-wise vector prediction task. Since iComFormer can only predict scalars, the only thing we can do is extend the prediction head from 1 to 3. When we rotate the whole coordinate system by R, the prediction target is changed accordingly to $\sum_{j \in N_i} R e_{ij} \in R^3$, but the input of iComFormer remains the same and the prediction will be the same. It can be seen that the directional information is lost in iComFormer for vector predictions, and by applying a group of Rs, we can obtain large errors. But for eComFormer, this can be naturally done by a single equivariant message updating, and the prediction will rotate accordingly to R.
> >
> > Tensor properties, including dielectric tensor, piezoelectric tensor, and elastic tensor, are of great interest to the material community and we believe the proposed eComFormer which achieves SOTA performances even for scalar predictions, is an important step towards solving the prediction of these properties (actually, no successful ML method has been proposed to predict even dielectric tensor with tensor order 2, due to the complex equivariant properties of this 3\*3 tensor). And we plan to tackle these challenges in the future work.
> >
> > Second, as demonstrated in the original paper, eComFormer outperforms iComformer in specific tasks, particularly those with a substantial number of training samples (e.g., over 60,000). This is because TP layers are more challenging to train and require a larger dataset to fully unleash their modeling capabilities. To better demonstrate this, we show the performance improvements of iComformer and eComformer in terms of the number of training samples and model config as follows. **It can be seen that when the number of training crystals increases from 44k to 60k, the performance improvement of eComFormer is much larger than iComFormer.** And eComFormer achieved the best result.
> > Additionally, eComFormer performs much better beyond iComFormer on Ehull in JARVIS, with similar configurations. It is worth noting that even when we use light-weight eComFormer configurations, with only k=16, and two node layers, eComFormer performs very close to iComformer on MatBench with MAE 16.8 meV/atom, but iComformer (16.5 meV/atom) uses k=25 and four node layers. We have not tuned the hyper-parameters heavily, and better performances of eComFormer are expected when using k=25 and three node layers.
> >
> > | Method | Number of training crystals | Test MAE (meV/atom) | Imp. | k | num. node layer|
> > | -- | -- | -- |-- |-- |-- |
> > |iComFormer|44296|27.2|-|25| 4|
> > |iComFormer|60000|18.26|8.94|25| 4|
> > |eComFormer|44296|28.4|-|25|3|
> > |eComFormer|60000|**18.16**|**10.24**|25|3|
> >
> > In conclusion, eComFormer not only shows superior performance in certain tasks and large-scale applications but also paves the way for breakthroughs in predicting complex material properties. Its design holds good promise for future developments and is likely to garner significant interest in the materials science community.
> >
> > The answers for question two are as follows.

---

> > ### Author Response · Authors · 2023-11-21
> > **Responses from authors for follow-up questions [Part 2]**
> >
> > 2. In addition, I have another question that I would like to discuss with the author. Given that I am not an expert in the field of materials, I am curious to know whether graphene, being a two-dimensional structure in the real world, encounters the issue of the lattice containing (0,0,0).
> >
> > Thank you for your question. To show that how graphene structure is represented in the real world, we show examples directly from JARVIS and the Materials Project. There are graphene structures with ids JVASP-14740 (https://www.ctcms.nist.gov/~knc6/static/JARVIS-DFT/JVASP-14740.xml), MP-48 (https://legacy.materialsproject.org/materials/mp-48/), MP-1040425 (https://legacy.materialsproject.org/materials/mp-1040425/), and MP-579909 (https://legacy.materialsproject.org/materials/mp-579909/). In fact this is how 2D graphene is modeled, either by putting it inside a 3D cell with large lattice constant along z direction (out of 2D graphene plane) to model the physical behavior of single 2D layer, or by stacking on more graphene layers separated by van der Waald distance (about 3.4 A between any two adjacent layers). So, regardless of whether a single 2D graphene layer is modeled or multiple stacked, the physical cell vector along out of plane is never zero.
> >
> > Yours sincerely, authors

---

> > > ### Comment · Reviewer_HGjX · 2023-11-22
> > >
> > > Thank you for your reply. I have no further questions about this work.
> > >
> > > Overall, the authors have proposed complete graph transformers for crystal material property prediction. This is an interesting study.
> > >
> > > I will improve my score.

---

> > > > ### Author Response · Authors · 2023-11-22
> > > > **Your suggestions helped us enhance the work**
> > > >
> > > > Dear reviewer HGjX,
> > > >
> > > > Thank you for your valuable feedback and constructive suggestions. Your insights have been instrumental in enhancing the clarity of our research. We are grateful for the time and effort you have dedicated to reviewing our work.
> > > >
> > > > Yours sincerely, authors

---

### Official Review · Reviewer_ZKHH · 2023-10-30

**Soundness:** 3 good
**Presentation:** 2 fair
**Contribution:** 3 good
**Rating:** 5
**Confidence:** 3

**Summary:**

Geometric graph representation learning for crystal material property prediction is challenging due to the periodic and infinite nature of crystals. This paper proposed SE(3) invariant and SO(3) equivariant crystal graph representations for geometric completeness of crystalline materials. Furthermore, two variants of a SE(3) transformer is proposed for crystalline material property prediction, i.e., iComFormer that uses invariant geometric descriptors of Euclidean distances and angles, and eComFormer that uses equivariant vector representations.

Experiments are conducted on three widely-used crystal benchmarks: JARVIS, the Materials Project, and MatBench. Experimental results showed that ComFormer achieves the best prediction accuracy across a range of crystal properties in most cases. The ablation study verifies the importance of geometric completeness of crystal graphs and higher rotation order features.

**Strengths:**

1. $\textbf{Problem Formulation}$: This paper identified an existing issue in previous related works, i.e., they map different crystal structures with different properties to the same graph representation and thus produce wrong property predictions. This is a good motivation. Based on this motivation, this paper aims to resolve this issue by proposing a new graph representation for crystalline materials to distinguish different crystals and remain the same under crystal passive symmetries.

2. $\textbf{Method}$: The proposed SE(3) invariant graph construction is sound and the subsequent crystal transformer networks are effective in crystal property prediction.

3. $\textbf{Experimental Results}$: The performance is superior than or comparable to the compared approaches across all three datasets in most cases.

**Weaknesses:**

1. $\textbf{Method}$

1.1 Is it just the way of lattice representation construction (Section 3.2 and Section 3.3) that introduces SE(3) invariant and SO(3) equivariant properties, or the proposed Transformer network (Section 4) has such properties? In other words, if the proposed Transformer network is applied to an ordinary graph (say a k-NN graph), does the SE(3) invariant and SO(3) equivariant properties still hold?

1.2 It seems that the node-wise transformer layer in SE(3) invariant message passing (Section 4.1) has no significant difference from the conventional graph Transformer (or graph attention network) [1]. Though detailed design may be different, but the general structure/scheme are similar. Correct me if I'm wrong.

1.3 It is unclear why edge feature $\textbf{e}_{ji}$ needs to be embedded using spherical harmonics, in second paragraph of Section 4.2. A justification is needed.

1.4 "$\textbf{TP}_0,\textbf{TP}_1,\textbf{TP}_2$ are tensor product layers" in the fourth paragraph of Section 4.2, What is tensor product layer and why need it here? Could you provide a concise and intuitive explanation?

2. $\textbf{Experiments}$

2.1 Is it possible to provide some qualitative experimental results (visualization) of SE(3) invariant property and/or SO(3) equivariant property, to prove the graph representation and the proposed Transformer network indeed learns such properties? It is not mandatory, but just out of curiosity.

2.2 In Table 5, the size of eComFormer is larger than iComFormer by two to three times. Any explanation or reason for this model size?

3. $\textbf{Writting}$

3.1 The paper spends a lot of space for background introduction and graph representation description, leading to just 1.5 pages for Transformer network design. It lacks the description of the rationale behind the network design, especially for SO(3) message passing, which causes a bit confusion in finding enough novelty of the proposed network.

$\newline$

[1] Veličković P, Cucurull G, Casanova A, Romero A, Liò P, Bengio Y. Graph Attention Networks. ICLR, 2018.

**Questions:**

Please kindly refer to the weakness part. I'd like to discuss with the authors for the above weaknesses part.

---

> ### Author Response · Authors · 2023-11-16
> **Responses from authors [Part 1]**
>
> Thank you for your recognition in terms of motivation, sound graph construction, and experimental results. We provide point-to-point responses for your mentioned questions as below.
>
> 1.  Is it just the way of lattice representation construction (Section 3.2 and Section 3.3) that introduces SE(3) invariant and SO(3) equivariant properties, or the proposed Transformer network (Section 4) have such properties? In other words, if the proposed Transformer network is applied to an ordinary graph (say a k-NN graph), do the SE(3) invariant and SO(3) equivariant properties still hold?
>
> Thank you for your question. The constructed graphs introduce SE(3) invariant and SO(3) equivariant properties, while the proposed Transformer networks maintain such properties. The iComformer and eComformer are specially designed for corresponding SE(3) invariant and SO(3) equivariant graphs, and cannot take ordinary graphs as input (ordinary graphs do not have that 3 lattice angles). The whole pipeline satisfies SE(3) invariance and they need to be used together.
>
> 2. It seems that the node-wise transformer layer in SE(3) invariant message passing (Section 4.1) has no significant difference from the conventional graph Transformer (or graph attention network) [1]. Though the detailed design may be different, the general structure/scheme is similar. Correct me if I'm wrong.
>
> We respectfully disagree with this. The major differences are as follows.
>
> First, GAT **only considers node features** and can not model 3D objects. The attention mechanism is $\alpha_{ij} = W(h_i, h_j)$, where $\alpha_{ij}$ is the attention weight for node pair i and j, and $h_i$ is the node feature for i.
>
> Second, according to $\alpha_{ij} = W(h_i, h_j)$, GAT **has node-wise attention** and can only model ordinary graphs with a single edge between nodes, while crystal graphs are multi-edge graphs.
>
> Different from GAT, SE(3) invariant message passing captures 3D information including Euclidean distances and lattice angles through edge-wise attention and is specifically designed for crystals.
>
> 3. It is unclear why edge feature needs to be embedded using spherical harmonics, in second paragraph of Section 4.2. A justification is needed. What is tensor product layer and why need it here? Could you provide a concise and intuitive explanation?
>
> We merge these two questions together since they are closely related. Let's begin with the features in SO(3) crystal graph and then discuss why TP layers are needed and why embed edge feature vectors into spherical harmonics. The edge features in SO(3) graph are $e_{ij} = p_j - p_i \in R^3$, which will change accordingly when you rotate the coordinate system by a rotation matrix. The aim of the proposed eComformer message passing will be, how we can make the best use of these $e_{ij}$ to capture rich geometric information without breaking SO(3) equivariance.
>
> To keep the equivariance property and finally obtain invariance predictions, there are limited operations that can be done to $e_{ij} = p_j - p_i \in R^3$. For example, one cannot use a nonlinear function like relu to $e_{ij}$. The limited operations one can use are norm(), scale by a scaler, or tensor product. However, the first two, norm() or scale, cannot capture angular information and only contain Euclidean distance information, while TP layers have the potential to capture this if properly designed like our proposed eComformer.
>
> Concretely, the Tensor product layer takes two inputs with different rotation orders, where a scalar belongs to rotation order 0, a 3D vector belongs to rotation order 1, and rotation order 2 vectors are of length 5. For example, when you use the tensor product with two 3D vectors, the operations have been done in TP($v_1$, $v_2$) will be, (1) calculate the tensor product $v_1 \times v_2$ with shape 3\*3, and then decompose this 3\*3 matrix into one scalar, one vector, and one 5D vector, through Clebsch–Gordan coefficients. The resulting scalar, vector, and high rotation order vector will rotate accordingly through Wigner_D conditioned on the rotation of the whole system (e.g. Wigner_D(R) for l=1 3D vectors will be exactly R). And more mathematical details can be found in [1].
>
> The reason to convert $e_{ij}$ to spherical harmonics is to feed it properly to TP layers, by normalizing the vector and converting it to one 3D vector with rotation order 1 (the direction of the vector) combined with a scalar with rotation order 0 (the length of the vector).
>
> Thus, we use Tensor product layers to process SO(3) equivariant crystal graphs, and prove that the design of eComformer considers all two-hop bond angle information in Appendix A.5. We also revised the paper accordingly in Sec. 4.2 to enhance the clarity.
>
> [1] Geiger, Mario, and Tess Smidt. "e3nn: Euclidean neural networks." arXiv preprint arXiv:2207.09453 (2022).

---

> ### Author Response · Authors · 2023-11-16
> **Responses from authors [Part 2]**
>
> 4. Is it possible to provide some qualitative experimental results (visualization) of SE(3) invariant property and/or SO(3) equivariant property, to prove the graph representation and the proposed Transformer network indeed learns such properties? It is not mandatory, but just out of curiosity.
>
> Sure, we provided further experimental verification of the SE(3) invariance of the proposed eComformer and iComformer for scalar properties. We use the MP dataset test set, and calculate the prediction differences of every test sample when the structure is rotated by R = [[0, -1, 0], [1, 0, 0], [0, 0, 1]], and found the differences before and after this rotation for all test samples are very small (less than 10^{-5} introduced by numerical errors). We further show that when a crystal structure ZnS (mp-560588) without inversion symmetry is reflected by R = [[-1, 0, 0], [0, -1, 0], [0, 0, -1]], the constructed graphs are totally different. For iComformer, the three lattice angles (cosine values) before inversion between atom 0 and atom 3 is (-0.8527, -0.8527, -0.1751) and is (0.8527, 0.8527, 0.1751) after inversion. For eComformer, the edge features $e_{ij}$ for atoms 0 and 3 before inversion is (3.8058, -2.1973,  0.7815) and changes to (-3.8058,  2.1973, -0.7815) after inversion, while lattice representation remain the same [-3.8058,0.,0.], [-1.9029,3.2959,0.], [ 0., 0., -6.2380], resulting in different SO(3) graphs.
>
> | MP test set prediction difference | Mean (applying rotation) |
> | -- | -- |
> |eComFormer|$1.25 * 10^{-6}$|
> |iComFormer|$2.05 * 10^{-6}$|
>
> Thus, iComformer and eComformer are SE(3) invariant when predicting scalars.
>
> 5.  In Table 5, the size of eComFormer is larger than iComFormer by two to three times. Any explanation or reason for this model size?
>
> The parameter numbers of eComformer mainly come from tensor product layers that are more expensive than normal message passings. We further conducted a stress test by decreasing the number of channels in equivariant tensor product layers by half, and the eComformer will only have 5.6 M parameters and is close to iComformer (4.1 M), the performance of the reduced eComformer is 29.1 meV/atom, which is very close to the full model (28.4 meV/atom), and still the SOTA performance beyond all previous works.
>
> 6.  The paper spends a lot of space on background introduction and graph representation description, leading to just 1.5 pages for Transformer network design. It lacks the description of the rationale behind the network design, especially for SO(3) message passing, which causes a bit confusion in finding enough novelty of the proposed network.
>
> Thank you for this suggestion, as discussed in the answer to question 3, The aim of the proposed eComformer message passing will be, how we can make the best use of these $e_{ij}$ to capture rich geometric information without breaking SO(3) equivariance during message passing.
>
> Following your suggestion, we **revised SO(3) message passing section and Appendix A.5 to emphasize these points and the design considerations**. We believe your suggestions enhanced the clarity of this paper.
>
> Yours sincerely, authors

---

> ### Author Response · Authors · 2023-11-21
> **A kind reminder**
>
> Dear Reviewer ZKHH,
>
> As the rebuttal period is approaching its end, we would be grateful if you could take a moment to review our responses. Your further insights will be immensely beneficial for us to improve our work and for the committee to make a well-informed decision.
>
> We understand that reviewing is a demanding task and truly appreciate the time and effort you have already invested in reviewing our work. In response to your valuable feedback, we have provided a detailed rebuttal addressing the questions and concerns raised.
>
> Please feel free to reach out if there are any additional clarifications or information you might need from our end.
>
> Thank you once again for your valuable time and consideration. We look forward to your response.
>
> Best regards, authors

---

> ### Author Response · Authors · 2023-11-22
> **A sincere reminder before the author-reviewer discussion deadline**
>
> Dear reviewer ZKHH,
>
> Since the deadline for the author-reviewer discussion is approaching soon, we would greatly appreciate it if you could provide your feedback within the remaining hours of the discussion period.
>
> Your response will be instrumental in helping us refine our submission and contribute more effectively to the field.
>
> We understand that you have a demanding schedule, and we appreciate the time and effort you dedicate to the review process. If there are any specific areas of our submission that you feel require further clarification or improvement, please do not hesitate to share your thoughts. We are fully committed to making the necessary revisions and providing additional information as needed.
>
> Yours sincerely, authors

---

### Official Review · Reviewer_v1jy · 2023-10-30

**Soundness:** 2 fair
**Presentation:** 2 fair
**Contribution:** 1 poor
**Rating:** 3
**Confidence:** 5

**Summary:**

The paper described an experimental approach to material property prediction by using graphs built on periodic crystal structures.

**Strengths:**

The authors should be highly praised for studying important real objects such as solid crystalline materials and for attempting to give formal Definitions 1-4. The paper is generally well-written and contains enough details that helped to understand the difficulties.

**Weaknesses:**

The word "problem" appears only once (in the abstract), though a rigorous and explicit problem statement might have helped to understand the unresolved challenges.

If we need a complete and invariant description of a periodic crystal, crystallographers solved this problem nearly 100 years ago by using Niggli's reduced cell of a lattice and then recording all atoms in so-called standard settings, see the book "TYPIX standardized data and crystal chemical characterization of inorganic structure types" by Parthé et al, which actually applies to all periodic crystals, not only inorganic.

However, all these standardizations have become obsolete in the new world of big and noisy data because the underlying lattice (not even a unit cell) of any periodic crystal is discontinuous under almost any perturbation, which is obvious already in dimension 1.

For example, the set Z of all integers is nearly identical to a periodic sequence with points 0, 1+ep_1, ..., m+ep_m in the unit cell [0,m+1] for any sma;; ep_1,...,ep_m close to 0, though their minimal periods (or unit cells) 1 and m+1 are arbitrarily different.

This discontinuity was reported for experimental crystals already in 1965, see Lawton SL, Jacobson RA. The reduced cell and its crystallographic applications. Ames Lab., Iowa State Univ. of Science and Tech.

A more recent example from Materials Project shows two nearly identical crystals whose unit cells differ by a factor of (approximately) 2
https://next-gen.materialsproject.org/materials/mp-568619
https://next-gen.materialsproject.org/materials/mp-568656

Moreover, atoms in any material always vibrate above absolute zero temperature, so their positions continuously change. As a result, any crystal structure with fixed atomic coordinates in a database is only a single snapshot of a potentially dynamic object, especially for proteins whose structures are also determined often by crystallization.

Hence the new essential requirement for any (better than the past) representations of crystals is a proved continuity under perturbations of atoms. This problem has been solved by Widdowson et al (NeurIPS 2022) with theoretical guarantees and practical demonstrations on the world's largest collection of materials: the Cambridge Structural Database. The underlying invariants have a near-linear time in the motif size and were used for property predictions by Ropers et all (DAMDID 2022) and by Balasingham et al (arxiv:2212.11246).

Despite the authors citing one of the papers above, Definition 1 didn't follow the definitions from the past that modeled a crystal as a periodic point set not as a graph.

Any graph representation of a crystal or a molecule is discontinuous because all chemical bonds are only abstract representations of inter-atomic interactions and depend on numerous thresholds on distances and angles, while atomic nuclei are real physical objects.

The traditional representation in (1) and Definition 5 (appendix A.3) of an isometry order atoms in a unit cell. While many crystals contain identical atoms, the invariance under permutations of atoms is missed because the paper has no words "permute" or "permutation".

The permutation invariance seems lost when using angles, which depend on point ordering.

The property prediction makes sense only for really validated data, not for random values, because neural networks with millions of parameters can predict (overfit) any data.

**Questions:**

The words "crystal graph" appear a few times before section 2.2 with references but without details. In Definition 1, could the authors please explain the meaning of "a crystal graph" and the symbols "=" and "->"?

Did the paper consider the invariance of crystal descriptors under permutations of atoms?

Does a manually chosen cutoff radius (line 3 from the bottom of page 4) make any invariant incomplete because perturbing atoms can scale a unit cell larger than any fixed radius?

Since the paper title promised complete transformers, have the authors found any geometric duplicates in the used datasets? Some duplicates were reported by Widdowson et al (MATCH 2022) as being investigated by several journals for data integrity.

The dataset details in section A.5 could include the most important information: whether crystals are simulated or experimental, and if the "ground-truth" properties are obtained from computer modeling or physical experiments.

It seems that all used crystals are hypothetical with simulated properties. Is the main contribution a computer program that predicts the outputs of other computer programs?

If initial property computations were too slow and most likely iterative, why not to stop their computations earlier instead of writing another program simulating the simulations?

How many hidden parameters and CPU hours were used for producing the experimental results?

What results in the paper are stronger than the past work below?

(1) continuity under perturbations in Theorem 4.2 and generic completeness of the density fingerprint in Theorem 5.1 from Edelsbrunner et al (SoCG 2021),
(2) continuity under perturbations in Theorem 4.3 and generic completeness with an explcit reconstruction in Theorem 4.4 from Widdowson et al (NeurIPS 2022),
(3) full completeness in Theorem 10 from Anosova et al (DGMM 2021), extended in arxiv:2205.15298,
(4) 200+ billion pairwise comparisons of all periodic crystals in the Cambridge Structural Database completed within two days on a desktop, see the conclusions in Widdowson et al.

Did the authors know about the classical results in crystallography cited above, starting from Niggli (1927), Lawton (1965), and Parthe (1987)?

---

> ### Author Response · Authors · 2023-11-16
> **Point-to-point responses to your questions [Part 1]**
>
> Thank you for your valuable time and comments, we provide point-to-point responses to your questions as follows.
>
> 1. The word "problem" appears only once (in the abstract), though a rigorous and explicit problem statement might have helped to understand the unresolved challenges.
>
> Thank you for your suggestion, we revised the introduction, related works, and appendix to include the discussions with [1-7], we believe your suggestion indeed makes the paper clearer for experts in crystallography. For the detailed problem definition and challenges, we provided them in Sec. 2.1 in the original paper.
>
> 2. If we need a complete and invariant description of a periodic crystal, crystallographers solved this problem nearly 100 years ago by using Niggli's reduced cell of a lattice and then recording all atoms in so-called standard settings.
>
> We kindly disagree with this. The discussion with Niggli's reduced cell of a lattice is included in the introduction. Niggli's reduced cell cannot be directly used to predict crystal properties, it just transforms a crystal structure to its standard form which still consists of coordinates, lattice vectors, and atom types.
>
> 3. However, all these standardizations have become obsolete in the new world of big and noisy data because the underlying lattice (not even a unit cell) of any periodic crystal is discontinuous under almost any perturbation. Moreover, atoms in any material always vibrate above absolute zero temperature, so their positions continuously change. As a result, any crystal structure with fixed atomic coordinates in a database is only a single snapshot of a potentially dynamic object. Hence the new essential requirement for any (better than the past) representations of crystals is a proven continuity under perturbations of atoms. This problem has been solved by Widdowson [1] with theoretical guarantees and practical demonstrations on the world's largest collection of materials: the Cambridge Structural Database. The underlying invariants have a near-linear time in the motif size and were used for property predictions by Ropers [7] and Balasingham [2]. Despite the authors citing one of the papers above, Definition 1 didn't follow the definitions from the past that modeled a crystal as a periodic point set not as a graph.
>
> We agree with this point. For crystal structures with changed positions, the property will change accordingly but not drastically.
>
> Hence, it is worth noting that the features used in our proposed graphs including Euclidean distances and two-hop angles, both are continuous under small perturbations. For the lattice, it is of great importance in determining various crystal properties as shown in a recent work [10]. All Euclidean distances, including lattice lengths, are encoded to 1/r. So for the lattice lengths a, b, c, which are usually large, changed to 2a, 4a, or 8a under perturbations, the maximum difference introduced in the representation after averaging is no larger than 1/ka (for k = 25 and a>4, this will be smaller than 0.01 which is of similar scale of a reasonable perturbation), and the proposed graph representations can capture these changes.
>
> Moreover, we **directly compared with the methods [2] and [7] you mentioned**, and we found that **the work [2] based on [1]** you mentioned actually uses graphs to represent crystals and have very close performances with CGCNN and **cannot achieve completeness** due to the usage of $k=12$ (to achieve completeness, $k>200$ is needed for the materials project according to [1]) while the performance of [7] is worse than CGCNN. We conduct this comparison mainly following [2] using experimentally verified crystals from the Materials Project-2018, with 30084 crystals. We also provided more discussions with [2] and [7] in Appendix 1.3.
>
> | Method | Number of training crystals | Test MAE |
> | -- | -- | -- |
> |CGCNN|29342|0.051|
> |PDD[1,2]|29342|0.047|
> |AMD [7]|24067|`0.78`|
> |iComformer|24067|**0.027**|
>
> The performance gaps between iComformer and you mentioned [1, 2, 7] are super clear even with fewer training samples ([2] did not release the code and dataset), with **more than 40\% decrease in MAE**. AMD [7] using the Gaussian Process needs more than 10 hours of training for 24067 samples and cannot scale to large datasets.
>
> Thus, your mentioned works [2] and [7] **do not achieve completeness when used as input to predict properties and have similar or worse performances with CGCNN, due to practical issues**. We included this comparison in the appendix and will release the dataset to make it easier for the ML community to follow the insightful works you mentioned [1,2,7].

---

> ### Author Response · Authors · 2023-11-16
> **Point-to-point responses to your questions [Part 2]**
>
> 4. Any graph representation of a crystal or a molecule is discontinuous because all chemical bonds are only abstract representations of inter-atomic interactions and depend on numerous thresholds on distances and angles, while atomic nuclei are real physical objects.
>
> We respectfully disagree with this. In the graph, we have not used chemical bonds but use Euclidean distances and 2-hop angles that are continuous.
>
> 5. The invariance under permutations of atoms is missed because the paper has no words "permute" or "permutation". The permutation invariance seems lost when using angles, which depend on point ordering. Did the paper consider the invariance of crystal descriptors under permutations of atoms?
>
> We respectfully disagree with this. To verify the permutation invariance of our proposed method, we randomly permute atom indexes in crystal structures from the Materials Project test set, and the predictions remain unchanged (the maximum property prediction difference before and after permutation is $2.24\*10^{-7}$ for Comformer without training, due to numerical error). This is because the graph construction process and message passing have not used any ordering information of atoms.
>
> 6. The property prediction makes sense only for really validated data, not for random values because neural networks with millions of parameters can predict (overfit) any data.
>
> We show that 30086 crystals in MP and 18865 crystals in Jarvis are experimentally validated (observed). We also further conducted an experiment on 30086 crystals (pure experimentally validated dataset) and compared with [2] and [7] you mentioned as shown above in question 3.
> For "because neural networks with millions of parameters can predict (overfit) any data", we kindly disagree, especially for the potential crystal space which is infinite. As far as we know, there is no evidence suggesting several million parameters can predict (fit) infinite crystal space.
>
> We assume you are talking about overfitting on the training set, but the robustness and prediction accuracy are verified on test sets of benchmarks, which are not seen by the trained models.
>
> 7. The words "crystal graph" appear a few times before section 2.2 with references but without details. In Definition 1, could the authors please explain the meaning of "a crystal graph" and the symbols "=" and "->"?
>
> We provided the detailed crystal graph construction process with figures in Appendix 1.1, and show the detailed crystal graph construction in Sec. 3.2 and 3.3. A crystal graph in our approach consists of node features $a_i$ (atom type) for every node, and edge features including Euclidean distances between node pairs and three lattice angles for the invariant graph, and a Euclidean vector for the equivariant graph. "=" means two crystal graphs have the same node features and corresponding edge features. "A->B" means if we have A, then B is true.
>
> 8. Does a manually chosen cutoff radius (line 3 from the bottom of page 4) make any invariant incomplete because perturbing atoms can scale a unit cell larger than any fixed radius?
>
> No. The proof of completeness is only based on the assumption that the constructed crystal graph is strongly connected. Actually, **no method can achieve completeness with a k (k nearest neighbors) that will make the crystal graph not strongly connected as shown in Appendix A.8.1, including PDD[1].**
>
> Further, to show the **robustness of our proposed crystal graphs** in achieving completeness, we use k=25 and show that it is enough to **fully reconstruct crystal structures with more than 700 atoms in a cell** (T2 dataset[8] used in [2] and [7]) as shown in the revised paper in Sec. 5.1.
>
> [1] Widdowson et al (NeurIPS 2022) -> Resolving the data ambiguity for periodic crystals. Daniel E Widdowson and Vitaliy Kurlin.
>
> [2] Balasingham et al (arxiv:2212.11246) -> Compact Graph Representation of crystal structures using Point-wise Distance Distributions. Jonathan Balasingham, Viktor Zamaraev, and Vitaliy Kurlin.
>
> [3] Recognizing Rigid Patterns of Unlabeled Point Clouds by Complete and Continuous Isometry Invariants with no False Negatives and no False Positives. Daniel E Widdowson and Vitaliy A Kurlin.
>
> [4] Widdowson et al (MATCH 2022) -> Daniel Widdowson, Marco M. Mosca, Angeles Pulido, Andrew I. Cooper, Vitaliy Kurlin.
>
> [5] Edelsbrunner et al (SoCG 2021) -> The Density Fingerprint of a Periodic Point Set. Herbert Edelsbrunner, Teresa Heiss, Vitaliy Kurlin, Philip Smith, and Mathijs Wintraecken.
>
> [6] Anosova et al (DGMM 2021) -> Algorithms for continuous metrics on periodic crystals. Olga Anosova and Vitaliy Kurlin.
>
> [7] Fast predictions of lattice energies by continuous isometry invariants of crystal structures. J Ropers, MM Mosca, O Anosova, Vitaliy Kurlin, AI Cooper
>
> [8] T2 dataset: Pulido, Angeles, et al. "Functional materials discovery using energy–structure–function maps." Nature 543.7647 (2017): 657-664.

---

> ### Author Response · Authors · 2023-11-16
> **Point-to-point responses to your questions [Part 3]**
>
> 9. Since the paper title promised complete transformers, have the authors found any geometric duplicates in the used datasets?
>
> The application of filtering duplicate crystals is not our focus. However, we do the following transformations to convert our proposed graphs to matrix form (Comformer-matrix) and do the screening. Following [1], we convert the crystal graph into a $n \times k \times 4$ matrix, where n is the number of atoms in a cell, k is the k-nearest neighbors, and 4 means we have 4 invariant features for every neighbor, including Euclidean distances, and 3 lattice angles. After this transformation, we can compute a $k*4$ invariant vector for this structure, and then use MAE between vectors to determine the structural differences between crystals (following AMD). By doing so, we screened the Jarvis dataset, and identified more than 100 duplicate pairs including ('JVASP-86097', 'JVASP-86697'), ('JVASP-21210', 'JVASP-1047'), ('JVASP-86322', 'JVASP-87063'), using threshold $10^{-6}$. The whole process, including graph construction, matrix transformation, and screening, takes less than 15 minutes.
>
> Additionally, contradicting to the reviewer’s claim, the two materials mentioned by the reviewer (mp-568619 and mp-568656), both of which are silicon carbide (SiC), are NOT duplicates. Rather, they are completely DIFFERENT, due to their different stacking faults or different sequences along the z direction, known as SiC polymorphs (https://en.wikipedia.org/wiki/Polymorphs_of_silicon_carbide). This commonly happens in SiC and many other close-packed metals and semiconductor crystals (e.g. titanium Ti, gallium arsenide GaAs, etc.), which are well-known in the materials science and semiconductor communities. Specifically, we can use the conventional notation of A, B, and C sites to denote three different projection locations of each vertical Si-C pair on the x-y plane (https://en.wikipedia.org/wiki/Close-packing_of_equal_spheres). “A” stands for the site (0,0), B for the site (1/3, 1/3), and C for the site (2/3, 2/3) in the fractional coordinates on the x-y plane.  Then, the stacking sequence for “mp-568619" is “CBACA” along the z direction, while the stacking sequence for “mp-568656" is “CBACABACAB”.  Although the first 5 stacking layers in the two structures are the same (CBACA), the rest 5 SiC stacking layers in “mp-568656" is “BACAB”, different from “CBACA”, which means one cannot by simply doubling the crystal structure of mp-568619 along z direction to obtain the exactly same structure as mp-568656. In fact, there are 3 “C”, 3 “B”, and 4 “A” stacking layers in mp-568656. Thus, **these two structures are completely different from each other**.
> Another clear evidence is their different electronic band gap: mp-568619 has a band gap of 1.75 eV, while the other one has a band gap of 2.01 eV, which otherwise, based on group theory, would have been the same if these two structures were simply related by doubling the structure. Additionally, the MAE error between two crystal structures calculated by ComFormer-matrix is 0.093 (>> $10^{-6}$), indicating they can be distinguished by our ComFormer model.
>
> It is worth noting that we are focusing on crystal property prediction, but not this task.
>
> 10. The dataset details in section A.5 could include the most important information: whether crystals are simulated or experimental, and if the "ground-truth" properties are obtained from computer modeling or physical experiments.
>
> Thank you for your suggestion, we added the information in the appendix A.5. 30086 crystals in MP and 18865 crystals in Jarvis are experimentally validated (observed). All crystal properties are DFT-calculated.
>
> 11. It seems that all used crystals are hypothetical with simulated properties. Is the main contribution a computer program that predicts the outputs of other computer programs?
>
> No, first, 30086 crystals in MP and 18865 crystals in Jarvis are experimentally validated (observed).
>
> Second, it is worth noting that in [2] and [7] suggested by the reviewer, only the MP dataset and the simulated lattice energy by force field calculation from another work [8] were used in their tests.
>
> Third, we disagree with "a computer program that predicts the outputs of other computer programs". The advances in the development of advanced electronic structure theory and exchange-correlation energy functional have significantly improved the accuracy of computational results, approaching the chemical accuracy of ~1 kcal/mol limited by experiments (including materials sample quality, inherent resolution of experimental techniques, and errors in measurements, etc.). The advanced electronic structure methods such as DFT become more and more reliable for property predictions with evaluated error shown in the JARVIS paper [9].
>
> [9]Choudhary, Kamal, et al. "The joint automated repository for various integrated simulations (JARVIS) for data-driven materials design." npj computational materials 6.1 (2020): 173.

---

> ### Author Response · Authors · 2023-11-16
> **Point-to-point responses to your questions [Part 4]**
>
> Continued answer for question 11: The advanced electronic structure methods such as DFT become more and more reliable for property predictions with evaluated error shown in the JARVIS paper [9]. However, their computational cost is high, which motivates many AI/ML works in the field of force field, including our proposed method towards accurate and fast prediction of materials properties and providing a short list of computationally screened potential candidates for experimentalists, thereby accelerating the discovery of new materials with desired properties.
>
> Last but not least, the method proposed here can be applied to both the computational dataset and experimental dataset if available.
>
> 12. If initial property computations were too slow and most likely iterative, why not stop their computations earlier instead of writing another program simulating the simulations?
>
> If the reviewer's question is about the curation of the dataset itself, the iterative calculations (more precisely self-consistent field calculations) are automatically stopped after the desired convergence is reached (e.g. when total energy difference or maximum residual atomic force is less than a given small threshold). Such iterative calculations need to be carefully done, otherwise, the calculated property will have large intrinsic errors. In other words, the initial calculations, though time-intensive, need to be carefully performed despite being slow, which is essential for generating accurate and high-fidelity data. These datasets such as MP and Jarvis serve as a robust foundation for training ML models, ensuring their predictive power and reliability (see the quality and accuracy addressed in the response to your question 11).
>
> If the reviewer's question is about materials modeling and simulations, it is worth noting that the physics, chemistry, and materials communities face concurrent challenges. But that's exactly why AI/ML approaches can be leveraged to advance the field of natural science. To be more specific, the materials world is complex and multiscale in nature [Handbook of Materials Modeling, S. Yip, Springer] which can span a huge length scale (10 orders of magnitude from Angstroms to meters) and a vast time scale (18 orders of magnitude from attoseconds to seconds/hours). DFT-based electronic structure calculations scale as $N^3$ where $N$ is the number of electrons/atoms for each electronic self-consistent field / iterative loop. So, for a single material/molecule, the computational cost of DFT calculations is reasonable and manageable, which can be typically done within a day or a few days, depending on (1) specific properties of interest (total energy/atomic forces, quasiparticle bandgap, exciton energy), (2) configuration setup (hybrid vs local/semilocal exchange-correlation energy functionals, whether or not considering spin-orbit coupling, etc.), and (3) computational resources and codes (CPU vs CPU+GPU parallelization, plane-wave basis set vs localized/polarization basis set vs dual basis set). However, it becomes very slow for materials/molecules with >1,000 atoms accompanied by the rapidly increased memory consumption. It becomes even worse for ab initio molecular dynamics calculations where very often thousands or more ionic steps are required for statistically meaningful physical properties, despite that such ab initio simulations are highly valuable for understanding and predicting many physical and chemical processes such as chemical reactions and catalyst design. By curating high-quality and consistent datasets of small systems and developing ML models such as iComformer and eComformer here, it will be possible to bridge both length-scale and time-scale bottlenecks mentioned in the beginning and advance the scientific understanding and predictions that are not possible with conventional approaches, as demonstrated in many recent works (e.g. Physical Review Letters 98, 146401 (2007); Physical Review Letters 104, 136403 (2010); Advanced Materials 31, 1902765 (2019); Nature Communications 11, 2654 (2020); Chemical Reviews 121, 10142 (2021)).
>
> 13. How many hidden parameters and CPU hours were used for producing the experimental results?
>
> As shown in the main paper, we use 4M params for iComformer. The training process is conducted on a single RTX A6000 GPU, where iComformer with 3 node layers takes 9.6 hours to train and eComformer 16 hours (much faster than previous SOTA ALIGNN).
>
> 14. Did the authors know about the classical results in crystallography cited above, starting from Niggli (1927), Lawton (1965), and Parthe (1987)?
>
> Thank you for your insightful query regarding our knowledge and consideration of seminal works in crystallography, specifically those by Niggli (1927), Lawton (1965), and Parthe (1987). We acknowledge the foundational importance of these studies in the field of crystallography and their various applications over the decades. (continued in Part 5).

---

> ### Author Response · Authors · 2023-11-16
> **Point-to-point responses to your questions [Part 5]**
>
> Below, we provide a brief overview of each work and its traditional applications, followed by an explanation of how our research diverges in its focus and objectives.
>
> - Niggli (1927): Paul Niggli’s contributions in 1927 laid the groundwork for modern crystallography through the development of the Niggli reduction algorithm. This algorithm standardizes crystal descriptions, making it easier to **compare different crystal structures**. Traditionally, Niggli's work has been pivotal in crystallographic data standardization, aiding in the systematic classification and comparison of crystalline materials.
> - Lawton (1965): Lawton's work in 1965 further extended the understanding of crystal symmetry and its practical applications. Lawton's contributions are particularly noted in the realms of inorganic compound charaterization, where understanding symmetry plays a crucial role in deciphering the physical and chemical properties of materials.
> - Parthe (1987): Parthe’s work in the late 1980s focused on the **systematic description of crystal structures**, contributing significantly to the field of structural crystallography. His methods have been extensively applied in the **identification and categorization of complex inorganic structures**. Parthe developed STRUCTURE TIDY code to help standardize crystal structure data and make it easier to identify identical or nearly identical structures by two standarization parameters \Gamma and CG. Optionally, this code can transform structures into Niggli's reduced cells and make it easy for **structural comparison** and finding out symmetry elements.
>
> While we recognize the importance and broad applications of these classical methods in crystallography, our research primarily focuses on the predictive modeling of material properties using advanced graph neural networks. These classical methods aimed at **structural standardization and comparison**. However, there is a huge gap from standardized (A, P, L) to crystal property prediction, which is exactly DFT algorithms and our proposed method want to resolve.
>
> In conclusion, while our research is informed by the rich history of crystallography, including the foundational work by Niggli, Lawton, and Parthe, it diverges in its application of computational methods for crystal property prediction. We believe that acknowledging these distinctions offers clarity on how our work contributes uniquely to the field.
>
> We hope this response adequately addresses your query and demonstrates the contextual understanding of our research within the broader landscape of crystallography.
>
> 15. What results in the paper are stronger than the past work below?
>
> (1) continuity under perturbations in Theorem 4.2 and generic completeness of the density fingerprint in Theorem 5.1 from Edelsbrunner et al (SoCG 2021),
> (2) continuity under perturbations in Theorem 4.3 and generic completeness with an explicit reconstruction in Theorem 4.4 from Widdowson et al (NeurIPS 2022), (3) full completeness in Theorem 10 from Anosova et al (DGMM 2021), extended in arxiv:2205.15298,
> (4) 200+ billion pairwise comparisons of all periodic crystals in the Cambridge Structural Database completed within two days on a desktop, see the conclusions in Widdowson et al.
>
> We respectfully disagree with this. The (1,2,3,4) you mentioned are designed to **identify the similarity of stable crystal structures with vector or matrix-based descriptors**. Our target is to predict the properties of crystal structures, which can be stable crystals or dynamic snapshots. The clear advantages of this paper beyond (1,2,3,4) are listed below.
>
> -> 1. These complete descriptors **can only be applied to stable crystal structures** and **do not consider atom types**. This makes them not suitable to do property predictions. For the property prediction task, we directly compared with [2] based on your mentioned (2,4), and found that when (2,4) is used to predict properties, it is no longer complete and uses graphs similar to CGCNN. While through extensive experiments, we have shown significant performance gaps beyond CGCNN, and **through direct comparison with [2], we show huge improvements (more than 40\% decrease in MAE)**.
>
> -> 2. Also, [2] based on (2, 4) **cannot achieve completeness** due to practical issues when doing property prediction. To achieve completeness of (2,4), the k, where k means k-nearest neighbors, needs to be predefined to be larger than the maximum number of atoms in the test set. This is against the logic of looking into the test set or assuming the largest cell the method will encounter. **For the T2 dataset[8], k needs to be larger than 700, which results in 700 * n edges to achieve completeness. However, as we show in the revised paper, our method can fully reconstruct crystal structures in the T2 test set by only using k=25**.
>
> continued in Part 6.

---

> ### Author Response · Authors · 2023-11-16
> **Point-to-point responses to your questions [Part 6]**
>
> Continued answer for question 15.
>
> -> Additionally, these descriptors cannot deal with dynamic crystal structures. For PDD (2,4), when two atom switches positions, **the crystal changes but PDD remains the same, and this case will occur in dynamic systems during trajectories**. For example, graphene and boron nitride (two famous 2D materials), both have hexagonal lattice, with close lattice constant of 2.46 A for graphene, and lattice constant of 2.54A.  If the lattice constants are changed to be the same under hypothetical dynamic states, then these two completely different crystals will have exactly the same PDD representations, and cannot be distinguished. **Our proposed method can still distinguish this case**, as we do not rely on any assumption about the stability of the crystal while proving completeness.
>
> -> Furthermore, these Euclidean distances-based descriptors **cannot distinguish chiral crystals with different crystal properties**.
>
> [10] Gong, Sheng, et al. "Examining graph neural networks for crystal structures: limitations and opportunities for capturing periodicity." Science Advances (2023)
>
> Yours sincerely, authors

---

> > ### Comment · Reviewer_v1jy · 2023-11-19
> >
> > Thank you for the update.
> >
> > > AMD and PDD ... cannot be directly used to predict crystal properties without breaking the completeness.
> >
> > How can you justify this claim?
> >
> > >There are several recent efforts, including AMD (Widdowson et al., 2022) and PDD (Widdowson & Kurlin, 2022) of matrix forms that are complete
> >
> > The latter paper has an example of periodic sequences that have the same AMD, hence AMD is not complete.
> >
> > >The completeness of AMD and PDD is based on the assumption that no two crystals can have the same structure but differ by a single atom type, which is only feasible for stable crystal structures
> >
> > There was no assumption. PDD was theoretically proved to be generically complete for all non-singular (but not all) crystals and PDD distinguished all periodic crystals in the world's largest database of real materials CSD (Cambridge Structural Database).
> >
> > The only found pairs with (almost) equal PDD within floating point errors were near-duplicates of the same structure (determined at different conditions such as temperature) or geometric duplicates that differed by atom types, which is physically impossible, hence all underlying papers are investigated by five journals for data integrity.
> >
> > What is your definition of a stable crystal structure?
> >
> > >our proposed graph representations can apply to dynamic crystal systems and are robust to various cell sizes
> >
> > Any graphs based on atoms as vertices and distance thresholds (cut-off radii) for edges are discontinuous. Is it really not obvious even for two points? If we fix a rule to connect any atoms at a distance at most d, then two atoms at the exact distance d are connected but shifting them away from each by any small amount removes this connection.
> >
> > If we connect any atom to a fixed number of nearest neighbors, say k=25, a crystal can have several atoms (say, 25,26) at the same distance. Even if we use an extra rule to break this tie, the neighbors can be swapped under tiny perturbation. Do you agree?
> >
> > > a sufficiently large number of neighbors k needs to be predetermined for any test crystal (must be larger than the number of atoms in the cell for any test crystal), which is not feasible
> >
> > If a number of considered neighbors is less than the number of atoms in a unit cell, how can you capture periodicity?
> >
> > If a unit cell has 3 atoms and we allow only connections to 2 nearest neighbors, these atoms can be connected into a triangular cycle without any knowledge about the crystal outside the cell. This example easily extends to any number of neighbors, say k=25.
> >
> > However, k=100 neighbors were enough to distinguish all real periodic crystals in the CSD.
> >
> > The "Invariant crystal graph construction" in section 3.2 uses indices of atoms and vectors (of a lattice basis), which makes the construction non-invariant under permutations of atoms and (more importantly) changes of a basis. Fixing any reduced basis, makes the construction discontinuous.
> >
> > Can you write down all steps from section 3.2 for a simple case of the rectangular lattice with the basis (4,0), (0,2) and two points in the cell, say (0,0) and (1,1), then actually prove that the proposed "invariant" is preserved by choosing *any other cell* for the same periodic set?
> >
> > Do the authors agree that any manually chosen cut-off radius automatically makes any invariant incomplete because a unit cell can be arbitrarily large, at least can become arbitrarily large under almost any tiny perturbation?
> >
> > Was the following example understood?
> >
> > The set Z of integers is nearly identical to a periodic sequence with points 0, 1+ep_1, ..., m+ep_m in the unit cell [0,m+1] for any small ep_1,...,ep_m close to 0, though their minimal periods (or unit cells) 1 and m+1 are arbitrarily different.
> >
> > >For the T2 dataset[8], k needs to be larger than 700
> >
> > No. If the papers on T2 crystals are more thoroughly read and the basic crystallography is understood, it would be clear that the distances to neighbors were computed only for atoms from an asymmetric unit (not the whole unit cell), which is one molecule of 46 atoms for T2 crystals.
> >
> > This asymmetric unit is usually smaller for most real crystals in the CSD.
> >
> > >Euclidean distances-based descriptors cannot distinguish chiral crystals with different crystal properties.
> >
> > What's your definition of chiral crystals and can you give an example of chiral crystals with different crystal properties?
> >
> > Since the authors honestly accepted that they didn't know about the classical results in crystallography, more time might be needed to understand these results.
> >
> > >Our target is to predict the properties of crystal structures
> >
> > Property prediction makes sense only when objects are already unambiguously represented. If one object has (infinitely many) different representations, the prediction should theoretically guarantee the same output on all inputs. If non-equivalent objects have the same representation but different properties, these properties cannot be predicted from the same input.

---

> > > ### Author Response · Authors · 2023-11-20
> > > **Point-to-point responses-2nd round [Part 1]**
> > >
> > > Dear reviewer, thank you very much for your valuable time and comments, we provide point-to-point responses to your questions.
> > >
> > > 1. AMD and PDD ... cannot be directly used to predict crystal properties without breaking the completeness. How can you justify this claim?
> > >
> > > Thank you for this question. We answered this question in the previous responses (first round) and we kindly point to the previous responses for question 3, 15, and Appendix A.1.3. Also, it is worth noting that **Comformer outperforms AMD and PDD more than 40\%** as shown in the previous responses for question 3.
> > >
> > > Limitations of PDD-ML. PDD-ML is more powerful than AMD-ML by converting the PDD matrix into crystal graphs like CGCNN-graph. By doing so, PDD-ML considers atom-type information. However, to achieve completeness for the PDD matrix, **a large enough k needs to be predetermined and fixed before training for any future test samples, where k needs to be larger than the maximum number of atoms in the cell for all test crystals.** Due to this practical concern, **PDD-ML only uses kmax = 25, which is far from completeness for their experiments on the Materials Project and T2 dataset.** Additionally, the **geometric information considered in PDD-ML for a fixed number of neighbors is the same as CGCNN with the same k as mentioned by the authors and verified by experiments in their paper.** As demonstrated by extensive experimental results, the proposed Comformer is significantly more effective than CGCNN in all tasks. Additionally, **by using k = 25 nearest neighbors, our proposed graph representations achieve completeness for the T2 dataset as shown in Tab. 8.**
> > >
> > > 2. AMD is not complete.
> > >
> > > Thank you for your information. We revised this sentence in our paper.
> > >
> > > 3. There was no assumption. PDD was theoretically proved to be generically complete for all non-singular (but not all) crystals and PDD distinguished all periodic crystals in the world's largest database of real materials CSD (Cambridge Structural Database).The only found pairs with (almost) equal PDD within floating point errors were near-duplicates of the same structure (determined at different conditions such as temperature) or geometric duplicates that differed by atom types. What is your definition of a stable crystal structure?
> > >
> > > Thank you for agreeing that **PDD cannot distinguish geometric duplicates that differed by atom types**.
> > >
> > > The stability of crystals means whether the crystal structure reaches the minimum energy configuration under enviromental conditions. For crystal structures that are not stable, a concrete example is non-equilibrium dynamic structures under external stimuli in experiment (e.g. excited and determined by ultrafast electron diffraction). Similarly, many atomistic configurations/structures are collected along the trajectories in classical/ab initio molecular dynamics simulations, or different configurations obtained from Monte Carlo sampling. As you mentioned in your question already, external temperature, pressure and electircal field, can change the crystal structures. In these non-equilibrium conditions, it is possible to have more distinct structures that may have same PDD matrix. Let's look at two examples. First, as we mentioned before, graphene and hexagonal boron nitride (two famous 2D materials), both have hexagonal lattice with close lattice constants (2.46 Angstroms for graphene, and 2.54 Angstroms for hexagonal boron nitride) and two A and B sites (A=(0,0) and B=(1/3, 1/3)). With a proper in-plane biaxial tensile strain applied onto graphene, its lattice constant can be exactly the same as boron nitride, then **can these two completely different crystals be distinguished by PDD?**. Second, even if the atom-type information were incorporated as additional feature to PDD, among the huge configuration space of unstable phases it is possible to have two enantiomers (i.e. left- vs right- chiral molecules/crystals). Each enantiomer is exactly a mirror image of the other one. In this case two enantiomers will have exactly the same set of chemical symbols/atom types, and the SAME PDD matrix due to the fact that PDD matrix is based on distance only. However, the atomic forces of two enantiomers will have OPPOSITIVE values normal to the the mirror plane, as long as they are perturbed away from the stable equilibrium structure. Note that, the effect on atomic forces is not the only difference. As shown on the webpage of Nobel Prize in Chemistry (https://www.nobelprize.org/prizes/chemistry/2001/press-release/), S-limonene and R-limonen are two enantiomers exactly related to each other by a mirror symmetry, but one smells as lemon, the other smells as orange. Another important difference is the optical activity - an ability of materials to rotate the plane of polarized light measured by polarimeter.

---

> > > > ### Author Response · Authors · 2023-11-20
> > > > **Point-to-point responses-2nd round [Part 2]**
> > > >
> > > > 4. If we connect any atom to a fixed number of nearest neighbors, say k=25, a crystal can have several atoms (say, 25,26) at the same distance. Even if we use an extra rule to break this tie, the neighbors can be swapped under tiny perturbation. Do you agree?
> > > >
> > > > First, it is worth noting the **PDD-ML work you mentioned with a fixed k will have this issue**. The work PDD matrix you mentioned, can tackle this issue only when it does not consider atom types. But PDD matrix, **cannot distinguish geometric duplicates that differed by atom types** as you mentioned above.
> > > >
> > > > Additionally, it is worth noting that we use the radius $r$ determined by the euclidean distance of the k-th neighbor, and then use this raidus to determine neighbors (the proposed graphs do not have a fixed number of neighbors, which is different from matrix form PDD). Furthuermore, to deal with the situation that "the neighbors can be swapped under tiny perturbation", one can choose a tolerance $t$ to your preferred tiny perturbations (e.g., <10^{-2}), and increase $r$ to $r + t$ if there are atoms between the sphere defined by raidus $r + 2t$ and $r$, and keep doing so until there is no atom between $r$ and $r+2t$. Then, the constructed graph will not have the issue you mentioned. Specifically, for perturbations that are too large (e.g. > t = 10^{-2}), the continuity concern may no longer be valid, and for perturbations that are small (e.g., <t=10^{-2}), they can be handled by using $r+t$.
> > > >
> > > > 5. If a number of considered neighbors is less than the number of atoms in a unit cell, how can you capture periodicity?
> > > >
> > > > Thank you for this question. The periodic information is captured by encoding lattice vectors as self-connecting edges as shown in Sec. 3.2 "Duplicates i1, i2, and i3 are included as neighbors of i to encode periodic patterns." Thus, as shown in the experimental part, Comformer graphs can reconstruct crystals from MP and T2 with just k=16 and k=25, which is **much smaller than the maximum number of atoms in a unit cell**.
> > > >
> > > > 6. The "Invariant crystal graph construction" in section 3.2 uses indices of atoms and vectors (of a lattice basis), which makes the construction non-invariant under permutations of atoms and (more importantly) changes of a basis. Can you write down all steps from section 3.2 for a simple case of the rectangular lattice with the basis (4,0), (0,2) and two points in the cell, say (0,0) and (1,1), then actually prove that the proposed "invariant" is preserved by choosing any other cell for the same periodic set?
> > > >
> > > > We kindly point the reviewer to the previous responses (first round) for question 5. **The experimental results already verified the permutation invariance.** Also, as shown in section 3.2, the lattice vectors are chosen based on invariant Euclidean distances and 2-hop angles.
> > > > For the proofs, we kindly point the reviewer to Appendix A.2.1, where we show that the proposed graphs satisfy periodic invariance, which is you mentioned "choosing any other cell for the same periodic set".
> > > >
> > > > 7. Do the authors agree that any manually chosen cut-off radius automatically makes any invariant incomplete because a unit cell can be arbitrarily large, at least can become arbitrarily large under almost any tiny perturbation? Was the following example understood? The set Z of integers is nearly identical to a periodic sequence with points 0, 1+ep_1, ..., m+ep_m in the unit cell [0,m+1] for any small ep_1,...,ep_m close to 0, though their minimal periods (or unit cells) 1 and m+1 are arbitrarily different.
> > > >
> > > > We kindly disagree, we kindly point the reviewer to the previous responses for question 8 about completeness, and in above question 5 about capturing periodic patterns. **The experimental verifications and proofs already suggest the opposite of this point of view.** For your mentioned example, the completeness of PDD for a fixed k will be broken (e.g., before the small perturbation, PDD uses k=100 and there are 100 atoms in the cell, and after the perturbation, there are 400 atoms in the cell and PDD with k =100 cannot achieve completeness.), but as we mentioned before, we **encode periodic patterns by encoding lattice vectors as self-connecting edges**, and the size of the cell will not directly influence the completeness of the proposed graphs. More specifically, as long as the k is large enough to form a strongly connected graph, the proposed graphs will be complete. And in which case k need to be increased is also discussed in Appendix A.8 (Actually, **no method can achieve completeness with a k (k nearest neighbors) that will make the crystal graph not strongly connected as shown in Appendix A.8.1, including PDD[1].**), but this case is not related to the increase of number of atoms in the cell.

---

> > > ### Author Response · Authors · 2023-11-20
> > > **Point-to-point responses-2nd round [Part 3]**
> > >
> > > 8. No. If the papers on T2 crystals are more thoroughly read and the basic crystallography is understood, it would be clear that the distances to neighbors were computed only for atoms from an asymmetric unit (not the whole unit cell), which is one molecule of 46 atoms for T2 crystals. This asymmetric unit is usually smaller for most real crystals in the CSD.
> > >
> > > Thank you for your information. Still, **PDD-ML you mentioned only uses k=25 and is not complete**. And the robustness in achieving completeness of our proposed method has been shown by using k=25 to reconstruct crystal structures in T2 as verified by experiments.
> > >
> > > 9. What's your definition of chiral crystals and can you give an example of chiral crystals with different crystal properties?
> > >
> > > Sure, we provide the detailed definition of chiral crystals and corresponding different crystal properties with examples as follows.
> > >
> > > Definition: Chiral crystals are a type of crystalline structure where the arrangement of atoms or molecules lacks inversion symmetry. This means that the crystal and its mirror image cannot be superimposed onto each other, similar to the relationship between left and right hands.
> > >
> > > First, let's begin with chirality for molecules. Two molecules that are chiral to each other can have dramatically different properties, for example, Thalidomide exists in two mirror-image forms: it is a racemic mixture of (R)- and (S)-enantiomers. The (R)-enantiomer has sedative effects, whereas the (S)-isomer is teratogenic. (For more information, please check https://www.acs.org/molecule-of-the-week/archive/t/thalidomide.html and https://en.wikipedia.org/wiki/Chirality )
> > > But **can PDD based on Euclidean distances distinguish these two small molecules?** The answer is no. These two molecules will have exactly the same PDD matrix.
> > >
> > > Second, for any crystal unit cell structure, if the motif within the unit cell lacks inversion symmetry, the crystal lacks inversion symmetry, and can have dramatically different properties. For example, Quartz (SiO₂) exists in two forms, left-handed (levorotatory) and right-handed (dextrorotatory) quartz, which are mirror images of each other.
> > > These two forms of quartz exhibit different optical properties. One form rotates the plane of polarized light to the left, and the other rotates it to the right. Additionally, some chiral crystals exhibit unique magnetic properties. Chiral magnetic crystals can show unusual arrangements of magnetic moments that lead to phenomena like the magneto-chiral effect, where the material's magnetic response is dependent on its chirality.
> > >
> > > 10. Since the authors honestly accepted that they didn't know about the classical results in crystallography, more time might be needed to understand these results.
> > >
> > > We did not say this. **We have already provided detailed explainations about the classical results in crystallography in answers for question 14**, and we kindly point the reviewer to the answers for question 14.
> > >
> > > 11. Property prediction makes sense only when objects are already unambiguously represented. If one object has (infinitely many) different representations, the prediction should theoretically guarantee the same output on all inputs. If non-equivalent objects have the same representation but different properties, these properties cannot be predicted from the same input.
> > >
> > > We kindly disagree with this point. We agree that when crystal structures are changed from A to B under perturbations, the property will change accordingly from v to v' but not drastically. But **we do not agree that when crystal structures are changed under perturbations (even tiny perturbations), the property will remain the same.(e.g. the system energy will always change for small perturbations of atom positions.)** Thus, what we aim to tackle is, how to enable the representation to capture any tiny structrual differences instead of mapping them to one identical representation, for better property predictions.

---

### Official Review · Reviewer_pQH7 · 2023-11-02

**Soundness:** 3 good
**Presentation:** 3 good
**Contribution:** 2 fair
**Rating:** 6
**Confidence:** 3

**Summary:**

This paper proposes a graph transformer network with rotation, translation, and periodic equivariance/invariance for learning on crystal graphs. It introduces a crystal graph representation with geometric completeness and equivariance. On top of it, the paper also proposes a transformer architecture with efficiency and expressivity. Experiments on three different benchmarks show the superiority of the proposed network.

**Strengths:**

- The paper introduces a crystal graph representation with equivariance and geometric completeness together with an equivariant transformer architecture, which tackles a practical problem in material analysis.
- The authors have provided adequate theoretical proof.
- The authors have presented extensive experiments on different benchmarks, efficiency analysis, and ablation studies, showing the effectiveness of the proposed network with superiority over prior works.
- The illustrations and explanations in the paper are clear and intuitive. I don't have any background in material science, but the problem statement in the paper sounds reasonable to me.

**Weaknesses:**

- I have limited knowledge about crystal graph learning, but the arguments in the paper look convincing to me. The authors mentioned learning higher-order properties in their own discussion of limitations. This could possibly be solved by the SO(3)-representation with sperical harmonic basis [1, 2].

[1] Thomas, Nathaniel, et al. "Tensor field networks: Rotation-and translation-equivariant neural networks for 3d point clouds." arXiv preprint arXiv:1802.08219 (2018).

[2] Fuchs, Fabian, et al. "Se (3)-transformers: 3d roto-translation equivariant attention networks." Advances in neural information processing systems 33 (2020): 1970-1981.

**Questions:**

- I don't have specific questions.

---

> ### Author Response · Authors · 2023-11-16
> **Responses from authors**
>
> Thank you for your recognition of our method in terms of tackling practical problems in material analysis, theoretical proofs, and extensive experiments.
>
> 1. The authors mentioned learning higher-order properties in their own discussion of limitations. This could possibly be solved by the SO(3)-representation with a spherical harmonic basis.
>
> Yes, we indeed plan to explore how to incorporate tensor field networks or TP layers to predict higher-order properties in crystals. These higher-order properties are of matrix form, for example, the dielectric tensors are rank two tensors and with shape 3\*3, while elastic tensors are rank 4 tensors with shape 3\*3\*3\*3 and are different from higher rotation order tensors generated from TFN. We plan to explore how to tackle these matrix form predictions in future works.
>
> Yours sincerely, authors

---

### Meta-Review · Area_Chair_Kvq9 · 2023-12-09

**Metareview:**

This submission proposes a graph transformer network with rotation, translation, and periodic equivariance/invariance for learning on crystal graphs.  Performance improvements are demonstrated empirically on a range of datasets showing consistent (small) improvement over baselines.  The reviewing was widely varied, with in particular reviewer v1jy being strongly opposed to the framework in general.  As this is building on a topic that is of established interest to the ICLR community, this review forum can serve as a reference for debate on the merits of these arguments, for which strong opinions have been stated on both sides.  We now turn to the other three reviews.  These were mixed with two weak accepts and a weak reject recommendation.  A common question for clarification was the motivation of the precise network architecture choices, while positives included the problem domain, formulation, and experimental results.  Release of an open source implementation at publication time would strongly improve reproducibility of the proposed method.

**Justification For Why Not Higher Score:**

There were some legitimate remaining reviewer questions about the motivation for certain aspects of network design.  Scores other than v1jy were leaning borderline.

**Justification For Why Not Lower Score:**

The submission is a sensible approach that is well described.

---

### Decision · Program_Chairs · 2024-01-16

Accept (poster)